# From Individual Experience to Collective Evidence:
# A Reporting-Based Framework for Identifying Systemic Harms

**Jessica Dai** [1]  **Paula Gradu** [1]  **Inioluwa Deborah Raji** [1]  **Benjamin Recht** [1]

## Abstract

When an individual reports a negative interaction with some system, how can their personal experience be contextualized within broader patterns of system behavior? We study the *reporting database* problem, where individual reports of adverse events arrive sequentially, and are aggregated over time. In this work, our goal is to identify whether there are subgroups—defined by any combination of relevant features—that are disproportionately likely to experience harmful interactions with the system. We formalize this problem as a sequential hypothesis test, and identify conditions on reporting behavior that are sufficient for making inferences about disparities in true rates of harm across subgroups. We show that algorithms for sequential hypothesis tests can be applied to this problem with a standard multiple testing correction. We then demonstrate our method on real-world datasets, including mortgage decisions and vaccine side effects; on each, our method (re-)identifies subgroups known to experience disproportionate harm using only a fraction of the data that was initially used to discover them.

## 1. Introduction

The impact of injustice is most acutely felt by the individual. But if an individual experiences harm, how can they know whether their experience is an isolated incident or part of a larger pattern of discrimination?

Fairness work has historically focused on model developers and third-party auditors as the main actors involved in creating fair mechanisms, motivating methods to construct models that are fair with respect to pre-defined subgroups at

development time (e.g., as surveyed in Pessach & Shmueli (2022))—or in identifying unfair ones, motivating post-hoc audits that occur after the entire decision-making process has completed (e.g., Byun et al. (2024); Martinez & Kirchner (2021)). However, in most applications where fairness is a concern, problems with the system may only emerge over time, and it is not necessarily obvious which subgroups might be important. Moreover, such approaches to fairness provide no mechanism for individuals to raise concerns.

It is exactly this question of individual agency that drives our work. In addition to normative reasons, which suggest that individuals ought to have a voice in expressing concerns with their treatment (e.g., the literature on contestability of algorithmic decisions (Vaccaro et al., 2019)), recent legislation has also highlighted individual reporting as a policy mandate for the governance of AI systems (e.g., the E.U. AI act (European Parliament, 2023)). While such legislation has yet to see full implementation, mechanisms for individual incident reporting already exist in a variety of domains, including consumer finance, medical devices, and vaccines and pharmaceuticals. A key component of reporting databases in the latter settings is that information from individual reports are aggregated to build collective knowledge about specific vaccines or pharmaceuticals—and, when applicable, this aggregated information can drive downstream decisionmaking, such as updating vaccine guidelines or drug treatment protocols (e.g., Oster et al. (2022)).

In this paper, we consider what a realistic approach to assessing fairness claims from an incident reporting database might look like in practice. We are primarily interested in designing a framework for the general public to report and contest large-scale harms by leveraging reports of *individual experience* to inform *collective evidence* of discrimination. To this end, we propose *reporting databases*, which allow individuals to submit reports of negative interactions, as a new mechanism for post-deployment fairness auditing. In particular, we identify conditions on reporting behavior and show how they can be used to to make inferences about rates of true harm in Section 3. Our formalization of the problem allows us to leverage known approaches to sequential hypothesis testing. In Section 4 we show how to instantiate two reasonable algorithms for our test and provide theoreti-

---
[1]Department of Electrical Engineering and Computer Science, U.C. Berkeley, Berkeley, CA, USA. Correspondence to: Jessica Dai <jessicadai@berkeley.edu>.

*Proceedings of the 42$^{nd}$ International Conference on Machine Learning*, Vancouver, Canada. PMLR 267, 2025. Copyright 2025 by the author(s).

cal guarantees for each. Finally, in Section 5, we illustrate the usefulness of our approach using real-world datasets, for applications with known disparity in per-subgroup rates of harm. On both real vaccine incident reports and on mortgage allocation decisions, our algorithm correctly identifies groups that disproportionately experience harm—and does so using a comparatively small number of reports.

## 1.1. Related work & application context

The reporting database problem is at the intersection of various challenges addressed in fairness and statistics. We discuss additional work, including application and policy context, in Appendix A.

**Algorithmic accountability via (individual) reports.** Some recent work considers methods for learning about fairness problems via individual reports from both theoretical (Globus-Harris et al., 2022) and practical (Agostini et al., 2024) perspectives. However, most discussion of individual experiences in machine learning fairness literature is limited to contexts where the objective is to assess, appeal, contest or seek recourse for that individual to change their *individual* outcomes, rather than forming a *collective* judgment about the system as a whole (Sharifi-Malvajerdi et al., 2019; Ustun et al., 2019; Karimi et al., 2022).

Work on identifying fairness-related issues via reporting data has typically focused on learning in batch contexts, e.g. via positive-unlabeled learning for handling disparate reporting rates across subgroups (e.g., Shanmugam et al. (2024); Wu & He (2022)). In other works, identifying disparate reporting rates is itself is the central challenge (e.g., Liu & Garg (2022); Liu et al. (2024)). On the other hand, an emerging body of literature from the human-computer interaction community develops the concept of *contestability* (e.g., Almada (2019); Vaccaro et al. (2019); Landau et al. (2024)); though contestability is still typically understood in terms of individual outcomes, we see our work as one possible path to implementing this ideal.

**Fairness auditing as hypothesis testing.** Existing proposals to formalize fairness auditing via hypothesis testing mainly consider batch settings (i.e. post-hoc or pre-deployment) (Cen & Alur, 2024; Cherian & Candès, 2023). Two more closely related works are that of Chugg et al. (2024) and Feng et al. (2024), who propose applying sequential hypothesis tests with the explicit goal of identifying problems in deployed systems in real time. However, as neither of these works study a reporting model, we propose fundamentally different tests: they test equality of means across different groups, while we compare within groups.

**Key definitions & clarifications.** Finally, we note that for AI systems, the term "incident database" has been used to

describe systems for monitoring the adverse impact of algorithmic deployments, which often take the form of accident catalogs that focus on one-off, large-scale events (e.g., Feffer et al. (2023); Raji et al. (2022); McGregor (2021); Ojewale et al. (2024); Turri & Dzombak (2023)). However, in the context of our work, we are actively excluding these accident catalog databases. Instead, we focus on reporting databases that provide records of individual experiences of adverse events that are tied to specific systems.

## 2. Model, Notation, and Preliminaries

The goal of constructing a reporting database is to determine whether some system that individuals interact with—for example, an (algorithmic) loan decision system, or a medical treatment—results in disproportionate harm to some meaningful subgroups. For the reporting database associated with a particular system, we will use $Y \in \{0, 1\}$ as an indicator variable that denotes the undesirable event corresponding to that system. For example, in loan decisions, this could correspond to the event that a highly-qualified individual was denied a loan; in the medical setting, this may be an adverse physical side effect due to the treatment.

**Subgroup definitions.** Individuals are characterized with feature vectors $X \in \mathcal{X}$, and we index individuals as $X_i$ ("features of individual $i$") or $X_t$ ("features of the individual who reports at time $t$"). Every individual $X_i$ "belongs to" at least one group $G$, and we will denote the event that $X_i$ belongs to $G$ as $\{X_i \in G\}$; we will use $\mathcal{G}$ to denote the set of all possible groups. This set of possible groups $\mathcal{G}$ can be defined arbitrarily as long as all groups can be determined as a function of covariates $\mathcal{X}$. We allow for groups to be overlapping—that is, we allow each individual $X_i$ to be in multiple groups so that $|\{G' \in \mathcal{G} : X_i \in G'\}| \geq 1$.

**Reference population.** The system for which the database is constructed naturally has a corresponding reference population of eligible individuals. For example, this could be everyone who has applied for a loan, or everyone who has been prescribed a certain medication. Thus, given a set of groups $\mathcal{G}$, we assume that it is possible to compute the composition of the reference population.

**Assumption 2.1** (Reference population). *For every $G \in \mathcal{G}$, the quantity $\mu_G^0 := \Pr[X \in G]$ is known. Throughout this work, we refer to the set $\{\mu_G^0\}_{G \in \mathcal{G}}$ as base preponderances.*

**Probabilistic model of reporting.** As the database administrator, the high-level goal is to determine whether there exists some subgroup $G \in \mathcal{G}$ where $\Pr[Y \mid X \in G]$ is abnormally high. Crucially, the database does not have access to information about every individual who interacts with the system; instead, individuals *may* report to the database if they believe that they experienced bad event $Y$. We thus let

$R_i$ be a random variable representing whether individual $i$ decides to report (with $R_i = 0$ indicating no report).

Each report $X_t$ is received sequentially, and assumed to be sampled i.i.d. from some underlying reporting distribution. Given a group $G$, we denote its corresponding mean among reports $\Pr[X_t \in G \mid R_t = 1]$ as $\mu_G$. We will sometimes refer to $\{\mu_G\}_{G \in \mathcal{G}}$ as (reporting) preponderances, as they represent the proportion of *reports* that each $G$ comprises. A central claim of this paper is that comparing $\mu_G$ to $\mu_G^0$—i.e., the extent to which group $G$ is (over)represented within the reporting database—can be a useful signal for $\Pr[Y \mid G]$ in a wide class of applications.[1]

## 3. Identifying Discrimination by Modeling Preponderance

A major challenge of assessing potentially-differential rates of harm across subgroups using only reporting data is to relate the event that someone submits a report to the event that they experienced harm. That is, if someone did experience a negative outcome, how likely is it for them to have reported it, and conversely, if someone submitted a report, how likely is it to reflect "true" harm? Moreover, as is known from prior work, reporting rates themselves can vary across subgroups.

Our central proposal is to conduct a hypothesis test for each group to determine whether it is overrepresented by a factor of $\beta$ among reports. That is, for each $G \in \mathcal{G}$, we test the following hypotheses:

$$\mathcal{H}_0^G : \mu_G < \beta \mu_G^0 \qquad \mathcal{H}_1^G : \mu_G > \beta \mu_G^0. \qquad (1)$$

In Section 4, we will discuss concrete algorithms for conducting this test sequentially and their corresponding theoretical guarantees. Before doing so, we first argue that testing for preponderance among reports, i.e., tracking $\mu_G$ in this way, can be a meaningful way to identify discrimination, even when exact reporting behavior is unknown. In Sections 3.1 and 3.2, we describe two distinct ways that this particular test can be interpreted; in Appendix B, we discuss some practical considerations for the modeling task.

### 3.1. Preponderance as relative risk

The first interpretation of our test allows us to make inferences about relative risk, the ratio between the rate of harm experienced by group $G$ and on average over the population. In this interpretation, the key quantity is the *report-to-incidence ratio*.

**Definition 3.1** (Report-to-incidence ratio). We define the *report-to-incidence-ratio (RIR)* as $\rho := \frac{\Pr[R=1]}{\Pr[Y=1]}$, and the

---

group-conditional analogue as $\rho_G := \frac{\Pr[R=1|G]}{\Pr[Y=1|G]}$.

[1]Because we allow groups to overlap, we cannot enforce $\sum_G \mu_G^0 = 1$ or $\sum_G \mu_G = 1$.

In Proposition 3.2, we show that if the group-conditional RIR of some group $G$ is at most some constant multiple of the population-wide RIR, then we can convert a lower bound on report preponderance into a lower bound on true relative risk (see Appendix C for proof).

**Proposition 3.2.** *Define the relative risk of group $G$ to be* $\mathrm{RR}_G := \frac{\Pr[Y=1|G]}{\Pr[Y=1]}$. *Suppose that for some group $G$ we have* $\rho_G \leq b \cdot \rho$. *Suppose that we determine that* $\mu_G \geq \beta \mu_G^0$ *for some* $\beta > 1$. *Then, the true relative risk experienced by $G$ is at least* $\mathrm{RR}_G \geq \beta/b$.

Suppose we take $\max_G \rho_G/\rho \leq b = 1.25$, i.e., no group over-reports 25% more often than the population average. Then, if a test identifies a group $G$ for which $\mu_G \geq 1.75 \cdot \mu_G^0$, this implies that the true relative risk for group $G$ is at least $\mathrm{RR}_G \geq 1.4$—that is, $G$ experiences harm 40% more frequently relative to the population average.

### 3.2. Preponderance as true incidence rate

We now discuss an alternate way to convert a lower bound on preponderance into a guarantee on real-world harm. In this case, we can infer the true incidence rate of harm (that is, no longer relative to the average) if we are able to estimate— or willing to make assumptions on—true and false reporting behavior in groups. Moreover, assumptions (or estimations) of these reporting rates need only be made in relation to the population average reporting rate $\Pr[R]$.

**Definition 3.3** (Reporting rates). Let $r := \Pr[R]$ be the average reporting rate over the full population. Let $\gamma_G^{\mathrm{TR}} := \frac{1}{r} \Pr[R_i = 1 \mid Y_i = 1, X_i \in G]$, $\gamma_G^{\mathrm{FR}} := \frac{1}{r} \Pr[R_i = 1 \mid Y_i = 0, X_i \in G]$. Finally, let $\mathrm{IR}_G := \Pr[Y \mid G]$ represent the true incidence rate, i.e. the likelihood that an individual in $G$ experiences $Y$.

Note that $r \cdot \gamma_G^{\mathrm{TR}}$ represents the (possibly group-conditional) rate at which individuals $X_i \in G$ who experience $Y$ actually report, while $r \cdot \gamma_G^{\mathrm{FR}}$ represents the rate that individuals $X_i \in G$ who do not experience $Y$ report. Thus, $\gamma_G^{\mathrm{TR}}$ and $\gamma_G^{\mathrm{FR}}$ represent how much more (or less) a particular group $G$ makes true or false reports relative to how much the whole population reports on average (which includes both true and false reports). The following proposition makes the relationships between $\gamma_G^{\mathrm{TR}}, \gamma_G^{\mathrm{FR}}$, and our quantity of interest $\mathrm{IR}_G$, more precise.

**Proposition 3.4.** *Suppose that, for some $G$, it is determined that $\mu_G \geq \beta \mu_G^0$ for some $\beta > 1$. As long as $\gamma_G^{TR} > \gamma_G^{FR}$ for every $G \in \mathcal{G}$,* $\mathrm{IR}_G \geq \frac{\beta - \gamma_G^{FR}}{\gamma_G^{TR} - \gamma_G^{FR}}$.

See Appendix C for the (short) proof. Proposition 3.4 shows that the exact computation of $\mathrm{IR}_G$ depends on reporting rates $\gamma_G^{\mathrm{TR}}$ and $\gamma_G^{\mathrm{FR}}$. While these quantities are not directly

estimable from reporting data—in fact, estimating reporting rates is itself a distinct research challenge (see, e.g., Liu et al. (2024))—these results can nevertheless guide qualitative interpretation of how severe $\text{IR}_G$ is.

For example, suppose a test is run for $\beta = 1.5$. Suppose $G$ overreports relative to the population average, with $\gamma_G^{\text{FR}} = 1$, and $\gamma_G^{\text{TR}} = 2$. Under these (generous) assumptions, we will have $\text{IR}_G = 0.5$, an extremely high incidence rate for any application—regardless of incidence rates for other groups.

Alternatively, suppose reporting rates did not vary by group (i.e., $\gamma_G^{\text{TR}} = \gamma^{\text{TR}}$ and $\gamma_G^{\text{FR}} = \gamma^{\text{FR}}$ for all $G$). Then, we can lower bound the disparities between true incidence rates across groups: if $G$ is flagged at $\beta > 1$, there must be some other group $G'$ with $\text{IR}_G - \text{IR}_{G'} \geq \frac{\beta-1}{\gamma^{\text{TR}} - \gamma^{\text{FR}}}$. If it is further assumed that $\gamma^{\text{FR}} = 0$, then $\text{IR}_G - \text{IR}_{G'} \geq \beta - 1$.

# 4. Identifying Subgroups with High Reporting Overrepresentation

How might the test proposed in Equation (1) be carried out in practice, with reports arriving over time, and what properties might we want for such a test? In this section, we provide two ways to instantiate this sequential hypothesis test. For each, we provide two types of guarantees. The first is (sequential) $\alpha$-validity, which, roughly speaking, guarantees correctness of groups identified in $\mathcal{G}^{\text{Flag}}$. More formally, we say that a sequential test is valid for a single group $G$ at level $\alpha$ if $\Pr[\exists t : \mathcal{H}_0^G \text{ rejected}] \leq \alpha$ when $\mathcal{H}_0^G$ is true. Because we are testing for all groups in $\mathcal{G}$ simultaneously, we say that a sequential test is valid with respect to all groups $\mathcal{G}$ if $\Pr[\exists t, \exists G : \mathcal{H}_0^G \text{ erroneously rejected}] \leq \alpha$.

The second type of guarantee is power, which guarantees that the test will identify a harmed group, if one exists. In particular, we are interested in the *stopping time* $T$ of the test, which is the number of samples required for the test to reject the first null, i.e. to raise an alarm for any group.

At a high level, our algorithms for conducting this test follow the protocol outlined in Algorithm 1. For each group $G$, we maintain a test statistic $\omega_t^G$ that is updated as reports $X_t$ are received over time. At each time $t$, each of these test statistics are compared to a threshold $\theta_t(\alpha)$, which depends on the test level $\alpha$; the null hypothesis $\mathcal{H}_0^G$ for group $G$ is rejected if $\omega_t^G > \theta_t(\alpha)$. For ease of exposition, Algorithm 1 is written so that groups corresponding to rejected nulls are collected in a set $\mathcal{G}^{\text{Flag}}$; in practice, a database administrator may choose to stop the test entirely as soon as one harmed group has been found.

Correcting for multiple hypothesis testing across groups is handled by a simple Bonferroni correction—that is, given a particular test level $\alpha$, we test each individual group $G$ at level $\alpha/|\mathcal{G}|$ rather than level $\alpha$. Though Bonferroni cor-

rections often seem onerous in non-sequential settings, we show that, for sequential problems, the Bonferroni correction incurs only a modest increase in stopping time.

In Section 4.1, we give a simple sequential Z-test-inspired approach which leverages a finite-time Law of the Iterated Logarithm. Section 4.2 presents a more complicated algorithm that leverages recent developments in anytime-valid inference. The main differences in each algorithm lie in how they implement Lines 1 and 6 of Algorithm 1—that is, how test statistics and thresholds are computed. For each instantiation of Algorithm 1, we show validity and power guarantees. Omitted proofs are given in Appendix C.

---

**Algorithm 1:** General protocol for testing overrepresentation

**Input:** Set of groups $\mathcal{G}$; base preponderances $\{\mu_G^0\}_{G \in \mathcal{G}}$; test level $\alpha$; relative strength $\beta$

1 Initialize test statistic $\omega_0^G$ for every $G \in \mathcal{G}$ and set threshold $\theta_0(\alpha)$;

2 Initialize set of rejected nulls (flagged groups) $\mathcal{G}^{\text{Flag}} := \emptyset$;

3 **for** $t = 1, 2, \ldots$ **do**

4     See report $X_t$;

5     **for** $G \in \mathcal{G}$ **do**

6         Update test statistic $\omega_t^G$ and compute threshold $\theta_t(\alpha)$;

7         **if** $\omega_t^G \geq \theta_t(\alpha)$ **then**

8             Add $G$ to $\mathcal{G}^{\text{Flag}}$ and take requisite action for $G$, if applicable.

---

### 4.1. Sequential Z-test

One simple observation that arises from the model presented in Section 2 is that if each report $X_t$ is drawn i.i.d. from some underlying distribution, then one might expect to be able to use concentration as a tool to conduct this test, since as time passes, the fraction of reports within the database from group $G$ should converge to the true mean $\mu_G$. We refer to this style of approach as a sequential Z-test, as it relies on measuring deviation from the mean.

**Updating the test statistic $\omega_t^G$.** Given this intuition, the test statistic is a simple count of the number of times a report from each group has been seen, i.e. (with $\omega_0^G = 0$),

$$\omega_t^G \leftarrow \omega_{t-1}^G + \mathbf{1}[X_t \in G]. \qquad (2)$$

**Setting the threshold $\theta_t(\alpha)$.** Given the way that $\omega_t^G$ accumulates evidence, one natural way to construct the threshold at each $t$ is to use the mean under the alternative, plus a correction term for both sample complexity and repeated testing over time. With $C$ set to either $\sqrt{\beta \mu_G^0 (1 - \beta \mu_G^0)}$ or

$1/2$, the threshold (including a Bonferroni correction) is

$$\theta_t(\alpha) := t\cdot\beta\mu_G^0 + C\sqrt{2.07t\ln\left(|\mathcal{G}|\frac{(2+\log_2(t))^2}{\alpha}\right)}. \quad (3)$$

**Theoretical guarantees.** Our first guarantee is a bound on the probability that any group is incorrectly flagged.

**Theorem 4.1** (Validity). *Running Algorithm 1 with $\theta_t(\alpha)$ as in Equation (3), setting $C = 1/2$, and $\omega_t^G$ updated as in Equation (2), guarantees that the probability that $\mathcal{G}^{Flag}$ will ever contain a group $G$ where $\mathcal{H}_0^G$ is true is at most $\alpha$, i.e.*

$$\Pr\left[\exists t : \exists G \in \mathcal{G}^{Flag} \text{ s.t. } \mathcal{H}_0^G \text{ holds}\right] \leq \alpha.$$

The choice of $C$ affects the nature of the guarantee: the true, finite-sample anytime-validity guarantee requires $C = 1/2$. If instead $C = \sqrt{\beta\mu_G^0(1-\beta\mu_G^0)}$, then, strictly speaking, the guarantee holds only asymptotically. However, a higher value of $C$ affects stopping time unfavorably, so the asymptotic approximation can be useful practically. In this case, care must be taken to ensure that the algorithm does not erroneously reject too early due to noise; one way to implement this is to mandate a minimum stopping time.

Finally, we give a stopping time guarantee for this test.

**Theorem 4.2** (Power). *Let $T$ be the stopping time of Algorithm 1 with $\theta_t(\alpha)$ as in Equation (3), $C = 1/2$, and $\omega_t^G$ as in Equation (2). Let $\Delta_{\max} = \max_{G\in\mathcal{G}}\mu_G - \beta\mu_G^0$. If $\Delta_{\max} > 0$, then $\Pr[T < \infty] = 1$. Furthermore, with probability $1 - \alpha/|\mathcal{G}|$, we have $T \leq \widetilde{\mathcal{O}}\left(\frac{\ln(|\mathcal{G}|)+\ln(1/\alpha)}{\Delta_{\max}^2}\right)$, and for any $\delta \in (0, \alpha/|\mathcal{G}|)$, we have with probability at least $1 - \delta$ that $T \leq \widetilde{\mathcal{O}}\left(\frac{\ln(1/\delta)}{\Delta_{\max}^2}\right)$.*

### 4.2. Betting-style approach

We refer to our second algorithm as a *betting-style* approach, due to the way we construct our test statistics (Shafer, 2021; Waudby-Smith & Ramdas, 2024; Vovk & Wang, 2021; Chugg et al., 2024); one way to interpret this approach is that the test "bets against" the null hypothesis $\mathcal{H}_0^G$ being true. We direct the reader to these references for more detailed technical exposition.

**Updating the test statistic $\omega_t^G$.** As in the previous approach, we let $\omega_t^G$ represent some accumulated amount of evidence against the null hypothesis $\mathcal{H}_0^G$ by time $t$, with a higher value of $\omega_t^G$ corresponding to greater level of evidence.[2] We initialize $\omega_0^G = 0$, and use the update rule

$$\omega_t^G \leftarrow \omega_{t-1}^G + \ln\left(1 + \lambda_t^G(\mathbf{1}_{X_t\in G} - \beta\mu_G^0)\right), \quad (4)$$

---

[2] The quantity $\exp(\omega_t^G)$ can also be referred to as an *e-value* (Vovk & Wang, 2021), a measure of evidence against a null hypothesis similar to a p-value.

with $\lambda_1^G, \ldots, \lambda_t^G \in [0, 1]$. Here, the algorithm accumulates a nonlinear function, with an adaptive parameter $\lambda_t^G$ that weights the influence of each new sample. Our setting of $\lambda_t$ is motivated by the goal of minimizing stopping time under the alternative, and thus to maximize $\omega_t^G$. Drawing from the well-studied problem of portfolio optimization in the online learning literature (Cover, 1991; Zinkevich, 2003; Hazan et al., 2016), we use Online Newton Step (Hazan et al., 2007; Cutkosky & Orabona, 2018) to ensure that $\omega_t^G$ is not too far from the best achievable in hindsight. This results in the following update for $\{\lambda_t\}_{t\geq 1}$:

$$\lambda_{t+1}^G \leftarrow \Proj_{[0,1]}\left(\lambda_t^G + \frac{2}{2-\ln(3)}\cdot\frac{z_t}{1+\sum_{s\in[t]}z_s^2}\right), \quad (5)$$

where $z_t = \frac{\mathbf{1}[X_t\in G] - \beta\mu_G^0}{1+\lambda_t^G(\mathbf{1}[X_t\in G] - \beta\mu_G^0)}$, and $\lambda_0 = 0$.[3]

**Setting the threshold $\theta_t(\alpha)$.** Unlike the sequential Z-test, we use the same threshold for all timesteps. Including a Bonferroni correction, we use $\theta_t(\alpha) := \ln(|\mathcal{G}|/\alpha)$ for all $t$; the motivation for this setting will become clear in our discussion of Theorem 4.3.

**Theoretical guarantees.** We first give a validity guarantee that is essentially identical to the Sequential Z-test.

**Theorem 4.3** (Validity). *Running Algorithm 1 with $\theta_t(\alpha) = \ln(|\mathcal{G}|/\alpha)$ and $\omega_t^G$ updated as per Equations (4) and (5) guarantees that the probability that $\mathcal{G}^{Flag}$ will ever contains a group $G$ where $\mathcal{H}_0^G$ is true is at most $\alpha$, i.e.*

$$\Pr\left[\exists t : \exists G \in \mathcal{G}^{Flag} \text{ s.t. } \mathcal{H}_0^G \text{ holds}\right] \leq \alpha.$$

This result follows directly from the prior work referenced at the beginning of this section. At a high level, every sequence $\{\exp(\omega_t^G)\}_{t\geq 1}$ is a non-negative super-martingale under $\mathcal{H}_0^G$; informally, this means that under the null hypothesis, the sequence $\{\exp(\omega_t^G)\}_{t\geq 1}$ should be non-increasing, in expectation. This allows us to apply Ville's inequality, which guarantees that it is unlikely that $\exp(\omega_t^G)$ ever becomes too large under $\mathcal{H}_0^G$. More specifically, for any $\alpha \in (0, 1)$, under the null, $\Pr[\exists t : \exp(\omega_t^G) > 1/\alpha] \leq \alpha$. Thus, maintaining a threshold of $\theta_t(\alpha) = \ln(|\mathcal{G}|/\alpha)$ is sufficient to provide a per-hypothesis $\alpha/|\mathcal{G}|$-validity guarantee, and thus $\alpha$-validity overall.

We also provide the following bound on stopping time; see Appendix C.3 for additional discussion of the $\omega_\star$ notion of gap.

**Theorem 4.4** (Power). *Let $T$ be the stopping time of Algorithm 1 with $\theta_t(\alpha) = \ln(|\mathcal{G}|/\alpha)$ and $\omega_t^G$ updated as per Equations (4) and (5). If $\max_{G\in\mathcal{G}}\mu_G - \beta\mu_G^0 > 0$, then, we*

---

[3] The constant $\frac{2}{2-\ln(3)}$ is due to Cutkosky & Orabona (2018), who give a tighter version of ONS than in Hazan et al. (2007).

*have that* $\Pr[T < \infty] = 1$. *Furthermore,*

$$\mathbb{E}[T] \leq \mathcal{O}\left(\frac{1}{\omega_\star^2} + \frac{\ln(|\mathcal{G}|) + \ln(1/\alpha)}{\omega_\star}\right)$$

*where* $\omega_\star := \max_{G \in \mathcal{G}, \lambda \in [0,1]} \mathbb{E}[\ln(1 + \lambda(\mathbf{1}_{X_t \in G} - \beta\mu_G^0))]$ *is the maximal expected one-step increase in* $\omega_t^G$ *over all groups and choices of* $\lambda$.

We conclude this section with two further remarks on Theorems 4.2 and 4.4 in the context of our test. First, our modeling in Section 3 measures severity of harm via a *multiplicative* factor of overrepresentation. However, both notions of gap in Theorems 4.2 and 4.4 also on the absolute size of the group $\mu_G$. Thus, for two groups $G$ and $G'$ with identical multiplicative gaps, i.e. $\mu_G/\mu_G^0 = \mu_{G'}/\mu_{G'}^0$, the test would stop faster in expectation for $G$ if and only if $\mu_G^0 > \mu_{G'}^0$. That is, if two groups are "harmed" to the same extent, both algorithms will identify the larger one first.

Second, for both tests, the Bonferroni correction results in only an additive factor ($\ln(|\mathcal{G}|)/\Delta_{\max}^2$ in Theorem 4.1, and $\ln(|\mathcal{G}|)/\omega_\star$ in Theorem 4.3) in stopping time over the scenario where we had only been testing the one group with the largest gap. This means that, in terms of worst-case guarantee on stopping time, the contribution of the Bonferroni correction is small relative to the contribution of the test level $\alpha$ and, especially, to the gap. In fact, the impact of Bonferroni on real-world data appears to be much smaller even than this additive term.

# 5. Real-World Examples

To demonstrate the applicability of our approach, we apply our framework to two real-world datasets. We begin by showing that our approach correctly and quickly identifies that young men experience myocarditis after the COVID-19 vaccine; then, on mortgage allocation data, we show that we identify known instances of discrimination under several reasonable reporting models. Code for all experiments, including instructions for data download and pre-processing, is available in the supplemental materials; additional experimental details can be found in Appendix D.

## 5.1. Myocarditis from COVID-19 vaccines

It is by now well-known that COVID-19 vaccines induce an elevated risk of myocarditis among young men. While initial suspicions of elevated myocarditis risk relied on case studies (e.g., Mouch et al. (2021); Larson et al. (2021); Marshall et al. (2021)), a more systematic understanding—including the pattern of disproportionate impact—was made possible by post-hoc analysis of reports from incident databases. Barda et al. (2021) appears to be the first analysis based on a database of reports, but did not disaggregate by demographic subgroups; the confirmation of young men as the most drastically-impacted group came in later studies (e.g., Witberg et al. (2021); Oster et al. (2022)).

In the U.S., these reports are collected inthe Vaccine Adverse Event Reporting System (VAERS). If we had been able to run the hypothesis tests proposed in the preceding sections on the reports collected in VAERS, would we have correctly identified this problem—and if so, how quickly? Concretely, we let $Y_i$ be the event that individual $i$ experiences myocarditis after receiving a COVID-19 vaccine, and run the test with the end-goal of identifying elevated incidence rate $\Pr[Y_i \mid X_i \in G]$ for group(s) $G$ corresponding to adolescent men (ages 12-17 and 18-29).

**Defining** $\mathcal{G}$**.** We consider (intersections of) sex and age buckets to be the subgroups of interest.[4] Age buckets are discretized into 0-4, 5-11, 12-17, 18-29, 30-39, 40-49, 50-64, 65-74, and 75+; the sex categories represented in the data are (binary) male and female. After removing groups for which no vaccines were recorded, $\mathcal{G}$ contains 29 groups.

**Setting** $\beta$**.** For this application, absolute incidence rate (that is, $\Pr[Y = 1 \mid G]$) is the quantity of interest to use for determining $\beta$. As suggested by Proposition 3.4, setting $\beta$ requires considering three quantities of interest: the threshold on an "unacceptable" incidence rate, the relative rates of true reporting $\gamma_G^{\mathrm{TR}}$, and the relative rates of false reporting $\gamma_G^{\mathrm{FR}}$. Then, we can set $\beta = \max_G \left((\gamma_G^{\mathrm{TR}} - \gamma_G^{\mathrm{FR}}) \cdot \mathrm{IR} + \gamma_G^{\mathrm{FR}}\right)$.

We will choose 0 as the threshold on an "unaccepptable" incidence rate. It is therefore sufficient to set $\beta = \max_G(\gamma_G^{\mathrm{FR}})$. While this is quantity cannot be determined from report data alone, a conservative assumption could be that any group erroneously reports at most twice the average reporting rate over the population, with $\gamma_G^{\mathrm{FR}} = 2.0$. If the algorithm is first run with $\beta = 2.0$, stopping and flagging a group very quickly, the test may be re-run with increasing values of $\beta$, as a higher $\beta$ corresponds to a more severe true incidence rate; thus, we also show results for $\beta = 2.5$ and $\beta = 3$.[5]

**Results.** We begin by running our algorithms on the actual sequence of reports in chronological order, as received in VAERS. In particular, we consider Algorithm 1 instantiated with $\omega_t^G$ updated according to Equation (2) and $\theta_t(\alpha)$ as in (3) and $C = 1/2$ (*Finite-sample Z-test*); with $\omega_t^G$ updated according to Equation (2) and $\theta_t(\alpha)$ as in (3) and $C = \sqrt{\beta\mu_G^0(1 - \beta\mu_G^0)}$ (*Asymptotic Z-test*); and with $\omega_t^G$ updated according to Equations (4) and (5), and $\theta_t(\alpha) =$

---

[4]While in principle it would have been interesting to also consider race/ethnicity, we are limited by the availability (and granularity) of the data given in VAERS, which does not include information on ethnicity/race in reports.

[5]Re-using this data is statistically valid due to the equivalence between one-sided hypothesis testing and confidence sequences.

|  | Asymptotic Z-test | | Finite-sample Z-test | | Betting-style test | |
|---|---|---|---|---|---|---|
|  | (M, 18-29) | (M, 12-17) | (M, 18-29) | (M, 12-17) | (M, 18-29) | (M, 12-17) |
| $\beta = 2.0$ | 34 (Feb. 22) | 256 (May 10) | 69 (Mar. 28) | 530 (May 30) | 61 (Mar. 23) | 241 (May 8) |
| $\beta = 2.5$ | 49 (Mar. 10) | 302 (May 15) | 74 (Mar. 31) | 546 (Jun. 1) | 69 (Mar. 28) | 259 (May 11) |
| $\beta = 3.0$ | 70 (Mar. 30) | 324 (May 18) | 111 (Apr. 20) | 612 (Jun. 6) | 80 (Apr. 5) | 302 (May 15) |

Table 1: On real historical sequence of myocarditis reports, time to identification of harmed groups. In each cell, we report the number of total reports to the rejection of the hypothesis corresponding to (M, 18-29) and the number of total reports corresponding to (M, 12-17). In all tests, the (M, 18-29) group is identified first—vaccines were authorized for the 12-15 age group only in May.

$\ln(|\mathcal{G}|/\alpha)$ (*Betting-style test*). For the asymptotically-valid Z-test, we require a minimum stopping time of $t = 25$, to prevent early rejections. We run all experiments for $\alpha = 0.1$.

In Table 1, we report the stopping time—that is, the number of reports it takes for the first null hypothesis to be rejected—of each algorithm for various values of $\beta$, as well as the corresponding date by which an alarm would have been triggered. Note that, in all tests, the (M, 18-19) group is identified first. This is consistent with the timeline of regulatory approvals: vaccines were authorized for ages 12-15 only by May 10 (Lovelace, 2021).

To explore the robustness of these results, we also run synthetic experiments, permuting the ordering of reports to get a sense of possible variance in the stopping time. We run 100 random permutations of the full set of reports. Figure 1 tracks the number of reports it takes for each algorithm to reject the null hypothesis for any group—that is, a scenario when the test is stopped and an alarm is raised as soon as one harmed group is identified. Each point on these plots reflects the number of trials (out of 100) in which a rejection has occurred by time $t$, when tests are run at $\beta = 2$.

In Figure 1, we compare the performance of the three algorithms. To interpret the figure, by time $t = 100$, the asymptotically-valid z-test had already identified harm in all 100 permutations; the betting-style test identified harm in around 80 permutations; and the finite-sample z-test had only identified harm in around 20 permutations. Figure 1 shows a clear ordering in terms of how quickly each algorithm tends to identify harm: the asymptotically-valid sequential z-test (dashed, red) is faster than the betting-style algorithm (solid, purple), which is faster than the finite-sample z-test (dotted, yellow).

Overall, our experimental results suggest that our proposed tests would in fact have been effective in determining that young men were disproportionately affected by myocarditis. Moreover, though it is difficult to determine exact timelines and the nature of clinical practice during early phases of the vaccine rollout, it is possible that such a test could have identified problems using less data—that is, more quickly—

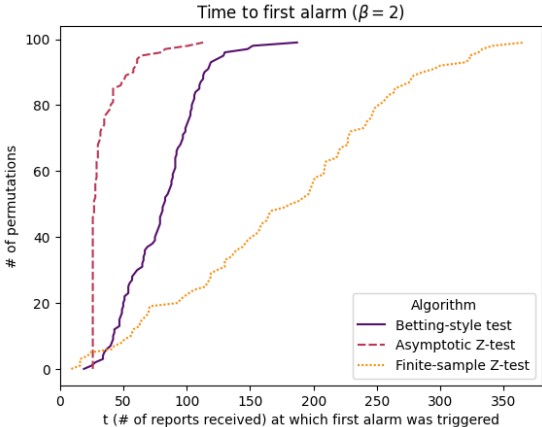

Figure 1: Stopping time (i.e. first identification of harm) for each algorithm, over 100 random permutations of COVID-19 vaccine reports, with $\beta = 2$. Each point on the plot reflects the number of trials (out of 100) in which a rejection has occured by time $t$.

than was actually used for this finding.

### 5.2. Mortgage Allocations

In 2021, Martinez & Kirchner (2021) found that, based on publicly-released data from the Home Mortgage Disclosure Act (HMDA), substantial racial disparities in 2019 loan approvals persisted even after controlling for financial status of applicants—most notably, healthy debt-to-income ratios (DTI). If loan applicants had been able to submit reports when they believed they had experienced unfavorable outcomes, could those reports have been used to identify this discrimination? If so, how accurately, and how quickly?

We are interested primarily in disparity among applicants with healthy DTI, even though all loan applicants would have been eligible to submit reports. Concretely, we let $A_i = 0$ be the event that a loan is not made to applicant $i$, and $Z_i = 1$ be the event that applicant $i$ has a healthy debt-to-income ratio. Then, we let $Y_i = \{A_i = 0, Z_i = 1\}$ be the event that individual $i$ has a healthy DTI and did not receive a loan, and run the test with the end-goal of identifying groups that have relatively high rates of loan denials for

| | Reporting model | Asymptotic Z-test | | Finite-sample Z-test | | Betting-style test | |
|---|---|---|---|---|---|---|---|
| | | Stopping time | Rel. risk | Stopping time | Rel. risk | Stopping time | Rel. risk |
| $\beta = 1.2$ | Correlated | 85 | 1.62 | 2002 | 1.67 | 638 | 1.70 |
| | All Denials | 69 | 1.59 | 1546 | 1.60 | 519 | 1.65 |
| | Anti-Corr. | 60 | 1.50 | 1065 | 1.53 | 403 | 1.65 |
| $\beta = 1.4$ | Correlated | 316 | 1.69 | 4306 | 1.73 | 1542 | 1.77 |
| | All Denials | 163 | 1.62 | 3214 | 1.72 | 1073 | 1.72 |
| | Anti-Corr. | 95 | 1.47 | 2215 | 1.66 | 718 | 1.68 |
| $\beta = 1.6$ | Correlated | 886 | 1.68 | 11755 | 1.73 | 4157 | 1.82 |
| | All Denials | 586 | 1.69 | 7410 | 1.72 | 2714 | 1.75 |
| | Anti-Corr. | 271 | 1.05 | 4668 | 1.72 | 1688 | 1.71 |
| $\beta = 1.8$ | Correlated | 4959 | 1.74 | —[1] | — | 16425[2] | 1.98 |
| | All Denials | 2703 | 1.72 | 29751[3] | 1.73 | 9977 | 1.89 |
| | Anti-Corr. | 935 | 1.58 | 14072 | 1.73 | 4629 | 1.76 |

Table 2: Average stopping times (i.e. time to first alarm) and true relative risk (i.e., $\frac{\Pr[A_i=0, Z_i=1 | X_i \in G]}{\Pr[A_i=0, Z_i=1]}$) of first-identified group over 100 random permutations, for varying $\beta$, across algorithms and reporting models. For $\beta = 1.8$, some combinations of algorithm/reporting model failed to stop within 40,000 steps for some trials: [1]stopped in 0/100 trials, [2]stopped in 99/100 trials, [3]stopped in 76/100 trials.

applicants with healthy DTI, i.e. $\frac{\Pr[A_i=0, Z_i=1 | X_i \in G]}{\Pr[A_i=0, Z_i=1]}$.

**Defining $\mathcal{G}$.** While Martinez & Kirchner (2021) analyzed disparities with respect to race, we define groups as all possible intersections of demographic features across gender, race, and age. The available race categories include Native, Asian, Black, Pacific Islander, White, and Latino; sex categories include female, male, and unknown/nonbinary; and age categories include <25, 25-34, 35-44, 45-54, 55-64, and 65+. In total, after removing groups which comprise less than 0.1% of all loan applicants, $\mathcal{G}$ contains 115 groups.

**Setting $\beta$.** In this application, the quantity of interest is relative risk, so we draw on Proposition 3.2 to inform our setting of $\beta$. We will set our relative risk threshold to be 1.2—that is, we want our algorithm to raise an alarm when any group experiences event $Y$ 20% more frequently than average over the population. Recall Definition 3.1 and Proposition 3.2: to flag relative risk at 1.2, $\beta$ should be set to $1.2 \cdot b$ where $b = \max_G \rho_G / \rho$, with $\rho_G = \frac{\Pr[R=1|G]}{\Pr[Y=1|G]}$ and $\rho = \frac{\Pr[R=1]}{\Pr[Y=1]}$; that is, $b$ is the extent to which the group-conditional report-to-incidence ratio for any group deviates from the population average report-to-incidence ratio.

As before, we can first test at $\beta = 1.2$, then re-test for higher values of $\beta$; in this case, we will also test $\beta = \{1.4, 1.6, 1.8\}$. Setting $\beta = 1.2$ corresponds to assuming $b = 1$, i.e., no variance in report-to-incidence ratios across groups; the additional values of $\beta$ suggest possible values of $b = 7/6$, $4/3$, and $3/2$, respectively.

**Reporting models.** The existence of verifiable disparities in this dataset allows us to evaluate the efficacy of our meth-

ods under varying models of reporting—that is, whether our algorithms identify groups that do in fact have high rates of healthy DTI denials, even if it is not the case that every report $X_i$ corresponds to $Y_i$ actually occurring. Modeling the idea that reporting behavior may be related to financial health, we simulate the following possible patterns of reporting.

(1) *Correlated:* The likelihood of reporting increases with financial health. That is, "Healthy" denials report with probability 0.9, "Manageable" with probability 0.5, "Unmanageable" with probability 0.3, and "Struggling" with probability 0.1. Under this reporting model, the 95th-percentile (among all groups) $\rho_G / \rho$ is 1.2, and $\max_G \rho_G / \rho = 1.4$.

(2) *All Denials:* All denials submit reports regardless of financial health. Under this reporting model, the 95th-percentile $\rho_G / \rho$ is 1.5, and $\max_G \rho_G / \rho = 2.3$.

(3) *Anti-Correlated:* The (unlikely) case where individuals with worse financial health are more likely to report, i.e. "Healthy" denials report with probability 0.1, "Manageable" with probability 0.5, "Unmanageable" with probability 0.7, and "Struggling" with probability 0.9. Under this reporting model, the 95th-percentile $\rho_G / \rho$ is 1.8, and $\max_G \rho_G / \rho = 2.7$.

Note that the ground-truth ratios $\rho_G / \rho$ would have been unknown at the time that a practitioner sets $\beta$; we are able to determine these only because we have full information about the dataset and control over the reporting model. However, these computations suggest that the assumptions on reporting rates implied by the settings of

$\beta = \{1.2, 1.4, 1.6, 1.8\}$ are generally reasonable, especially after considering outliers—note the disparity between the 95th-percentile vs max ratios of $\rho_G/\rho$, especially for the *All Denials* and *Anti-Correlated* models.

**Results.**   We run all three algorithms discussed in Section 4 at $\alpha = 0.1$, for all four reporting models discussed above, and for $\beta = \{1.2, 1.4, 1.6, 1.8\}$. For the asymptotically-valid Z-test, we (heuristically) choose a higher minimum stopping time of 50, to reflect the more challenging problem instance compared to the vaccine reporting problem. For each algorithm, reporting model, and $\beta$, we again run 100 random permutations.[6]

One important question for this application is the extent to which our tests identify the type of harm we are interested in, across various reporting models: while the algorithms guarantee statistical validity in terms of overrepresentation (i.e., in terms of whether $\mu_G \geq \beta \mu_G^0$), they cannot intrinsically guarantee that reports themselves reflect true harm. With the benefit of hindsight (and access to the full dataset), we are able to calculate "ground truth" relative risks; the hope for our algorithms is that they identify groups that actually do experience elevated relative risk.

Our results suggest that this is indeed generally the case, although the actual behavior varies by algorithm and reporting model. Table 2 shows report the average stopping times and average true relative risks of the first-identified group for 100 permutations. Across all algorithms and values of $\beta$, the stopping time under the *Correlated* reporting model is the longest, followed by the *All Denials* and *Anti-Correlated* reporting models. On the other hand, the relative risk of the group that is first identified in each of these settings follows the same ordering, with the *Correlated* model having the highest relative risk. That is, more "favorable" reporting behavior required a test to run longer, but the group identified is more severely harmed, whereas more "adversarial" reporting behavior raised an alarm sooner, but identified a less severely-harmed group.

Similar tradeoffs arise when comparing algorithms: the asymptotically-valid Z-test stops far more quickly, but appears to identify less severely-harmed groups. On the other hand, while the betting-style test and the finite-sample Z-test tend to identify similarly-harmed groups, the latter stops much faster than the former; overall, it appears that the betting-style test is a reasonable approach to balancing fast identification with confidence in the severity of harm.

While overall trends across algorithms and reporting models are consistent across values of $\beta$, seeing these results for different $\beta$ highlights an additional insight. While it is to be

expected that stopping times (and the ground-truth relative risks) should increase with $\beta$, the increase in stopping time is dramatic—by sometimes by orders of magnitude—even for what appear to be relatively small changes in $\beta$. Moreover, the *disparity* in stopping time across reporting models also increases dramatically with $\beta$. In fact, for $\beta = 1.8$, some combinations of reporting and algorithm do not stop within 40,000 steps in at least one trial.

## 6. Discussion

This work is an initial approach to using reporting databases for post-deployment auditing; we believe there is a rich range of future work that develops the ideas in this paper, both technically and conceptually.

On the statistical and algorithmic side, because our framework allows for plugging in any existing sequential test, new methods that control for multiple hypothesis testing both over time and over the number of distinct hypotheses would be directly beneficial for this application. On the other hand, one might hope for online methods that do not require pre-specifying hypotheses and instead develops them sequentially in a quasi-unsupervised fashion, or that improve guarantees by exploiting relationships across hypotheses, as has proven useful in multi-objective learning.

More conceptually, while the application examples in Section 5 are somewhat stylized, they demonstrate that reporting databases can be promising starting points for new types of post-deployment evaluation. For reporting databases to be practically useful, there are a plethora of additional considerations to incorporate from a variety of disciplines. For instance, if a reporting system was available, how would individuals engage with them in theory, and in practice? To what extent do, and should, individual incentives affect the database, and how it is designed? How can the result of a test (a null hypothesis rejection) be contextualized by existing and emerging legal frameworks?

To the best of our knowledge, we are the first to propose individual incident reporting to identify patterns of disproportionate harm in interactions with a particular system; more generally, however, one might imagine that similar reporting systems can be developed to provide insights about concerns beyond fairness. In fact, while the framework introduced in our work is not intrinsically about algorithmic deployments, it is one way to operationalize recent regulatory movement in AI policy towards allowing for or requiring individual reports. Any way to make such reports actionable at large scale must, to some extent, aggregate of individual reports to develop more systematic evaluations of an underlying algorithm. We therefore see our work as one step towards giving voice to individual experiences—and towards having them make a difference.

---

[6]Since we are simulating reporting, there is no "true" historical sequence of reports to run an algorithm on, unlike in Table 1.

## Acknowledgements

We are grateful to Ian Waudby-Smith, Kevin Jamieson, and Robert Nowak for helpful discussions in developing this work.

JD is supported in part by the National Science Foundation Graduate Research Fellowship Program under Grant No. DGE 2146752. Any opinions, findings, and conclusions or recommendations expressed in this material are those of the author(s) and do not necessarily reflect the views of the National Science Foundation. JD also thanks the AI Policy Hub at UC Berkeley for funding support in the 2023-2024 academic year. BR is generously supported in part by NSF CIF award 2326498, NSF IIS Award 2331881, and ONR Award N00014-24-1-2531. IDR is supported by the Mozilla Foundation and MacArthur Foundation.

## Impact Statement

This work proposes individual reporting databases as a promising approach to post-deployment evaluation for fairness and performance. While we hope that the work impacts the practice of post-deployment evaluation, we do not believe its societal impacts merit discussion beyond what is already given in the paper and its appendices.

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

## A. Additional Related Work

**Identifying and defining subgroups.** One approach to subgroup definition, following the line of work in multicalibration (Hébert-Johnson et al., 2018), is to simply enumerate over all possible combinations of covariates. For sequential problems, per-group guarantees can be provided for subgroups that are learned online (Dai et al., 2024), though these guarantees are in terms of prediction quality rather than statistical validity. For sequential experiments, Adam et al. (2024) propose an approach to early stopping that does not require the experimenter to pre-specify the group experiencing harm, but instead identifies those who appear to be harmed more frequently. Though this is in spirit similar to the idea of identifying groups that report more frequently, their algorithm is substantially different from ours, in addition to the distinct application context.

**Sequential and multiple hypothesis testing with anytime guarantees.** One of our proposed tests provides anytime-validity guarantees by adapting the analysis of (Jamieson et al., 2014) and (Balsubramani, 2014). Our second proposed test leverages the recent literature on e-values (e.g. (Waudby-Smith & Ramdas, 2024; Vovk & Wang, 2021)), which can be used to construct sequential tests that have validity guarantees in finite samples. While existing literature suggests methods for global null testing that can aggregate e-processes (e.g., Cho et al. (2024) or Chi et al. (2022)), such approaches are unable to provide per-hypothesis guarantees. More classical approaches include Wald's Sequential Probability Ratio Test (SPRT) and its extensions, such as Max-SPRT (Kulldorff et al., 2011), or a sequential generalization of the Holm procedure Bartroff & Song (2014).

**Application & policy context.** Sequential hypothesis tests have been used for real-world monitoring of adverse incidents in vaccines and medical devices (see, e.g., (Shimabukuro et al., 2015)). Descriptive studies have identified disparate adverse impacts in pharmaceutical (Lee et al., 2023; Whitley & Lindsey, 2009) and vaccine settings (Oster et al., 2022). More generally, post-market surveillance is standard across various industries, especially as it relates to product safety enforcement and monitoring.

In AI policy contexts, there have already been several calls to adopt a post-market surveillance regime for AI governance (e.g., Raji et al. (2022)). The U.N. General Assembly's first AI Resolution (7 8/265 and 78/311) explicitly encourages "the incorporation of feedback mechanisms to allow evidence-based discovery and reporting by end-users and third parties of [...] misuses of artificial intelligence systems and artificial intelligence incidents" (Assembly, 2024). In the U.S., Biden's (now repealed) AI Executive order explicitly directs the Department of Health and Human Services (HHS) to "establish a [...] central tracking repository for associated incidents that cause harm, including through bias or discrimination" (Biden, 2023). In the E.U., Chapter IX of the 2024 EU AI Act focuses on post-market surveillance, with Articles 85 and 87 specifically highlighting individual reporting of harms.

## B. Practical Considerations

**Choosing $\mathcal{G}$.** In our experiments in Section 5, we choose to define subgroups as all possible combinations of available demographic characteristics. That said, a practitioner may seek to define $\mathcal{G}$ more carefully in accordance with their application. For instance, if the goal is to illustrate discrimination in a legal sense, $\mathcal{G}$ should be defined with respect to (protected) demographic features, rather than arbitrary combinations of covariates. On the other hand, groups need not be solely demographic, which allows our approach to test for safety rather than solely fairness. For example, $\mathcal{G}$ could include which batch of a medication an individual received; our tests could then help identify whether some batches were improperly manufactured.

**Baseline rates $\{\mu_G^0\}_{G \in \mathcal{G}}$.** A natural question that arises from the modeling in this section is how $\{\mu_G^0\}_{G \in \mathcal{G}}$ can be determined, or if Assumption 2.1 is strictly necessary. Practically speaking, these base preponderances may be estimated, possibly with some amount of noise; however, the estimation problem can be addressed with standard techniques and is not core to our contribution. Similarly, in practice these baseline preponderances may change over time (e.g. if some subgroups increased uptake of a vaccine, or applied for loans more frequently, over time); however, such situations are relatively straightforward to handle under our algorithmic frameworks (see, e.g., the variants discussed in Chugg et al. (2024)). We therefore focus on the case where we have access to the true, underlying values of $\{\mu_G^0\}_{G \in \mathcal{G}}$ for ease and clarity of exposition.

Note that testing against base preponderances of the reference population (i.e., to compare $\mu_G$ to $\mu_G^0$) is a new test proposed by this work, and the analysis in Sections 3.1 and 3.2 is specific to this test. Existing approaches to monitoring in incident

databases compare to different baselines, most commonly the historical overall incidence rate for the specific symptom, sometimes by subgroup (Shimabukuro et al., 2015; Kulldorff et al., 2011; Oster et al., 2022). These baselines could, in principle, be plugged into the algorithms in Section 4, but new analysis for (possibly group-varying) reporting rates would be necessary to draw inferences about analogous quantities of interest (e.g., RR or IR), as current approaches do not generally consider reporting behavior. In contrast, our modeling allows us to identify what quantities may affect the true incidence rate even if they may be unmeasurable.

**Setting $\beta$.** Finally, to run the test proposed in Equation (1), it is necessary to determine how to set the value of $\beta$. As we will see in Section 4, when $\beta$ is set too high, then the test may never identify problematic groups, or identify them more slowly; on the other hand, as is clear from the previous subsections, rejecting the null hypothesis for a smaller $\beta$ requires more stringent assumptions on reporting behavior. Thus, we suggest a procedure to set $\beta$ as follows: (1) choose a relative risk or incidence rate threshold where it would be problematic for any group if $\mathrm{RR}_G$ or $\mathrm{IR}_G$ surpassed that threshold; (2) make the corresponding assumptions about reporting behavior; (3) use these quantities to compute a reasonable $\beta$. We give some example computations in Section 5. Due to an equivalence between hypothesis testing and confidence intervals, it is statistically valid to rerun tests with different $\beta$s once data collection has begun. Thus, it may be prudent to begin by setting the lowest $\beta$ that reporting assumptions would allow; then, if the tests appear to be stopping very quickly, to re-run them at higher $\beta$s, which would allow a practitioner to get a better sense of the severity of the harm.

## C. Omitted Proofs

### C.1. Omitted proofs from Section 2

We prove Proposition 3.2, restated below.

**Proposition 3.2.** *Define the relative risk of group $G$ to be $\mathrm{RR}_G := \frac{\Pr[Y=1|G]}{\Pr[Y=1]}$. Suppose that for some group $G$ we have $\rho_G \leq b \cdot \rho$. Suppose that we determine that $\mu_G \geq \beta \mu_G^0$ for some $\beta > 1$. Then, the true relative risk experienced by $G$ is at least $\mathrm{RR}_G \geq \beta/b$.*

*Proof of Proposition 3.2.* First, note that by definition of $\rho$, $\rho_G$, and $\mathrm{RR}_G$, we have

$$\rho_G \leq b \cdot \rho \iff \frac{\Pr[R=1 \mid G]}{\Pr[Y=1 \mid G]} \leq b \cdot \frac{\Pr[R=1]}{\Pr[Y=1]} \iff \mathrm{RR}_G \geq \frac{\Pr[R=1 \mid G]}{\Pr[R=1]} \cdot \frac{1}{b}.$$

By Bayes' rule, $\frac{\Pr[R=1|G]}{\Pr[R=1]} = \frac{\Pr[G|R=1]}{\Pr[G]} = \frac{\mu_G}{\mu_G^0}$; furthermore, by assumption, we have $\frac{\mu_G}{\mu_G^0} \geq \beta$. The result follows from combining with the previous display. □

We prove Proposition 3.4, restated below.

**Proposition 3.4.** *Suppose that, for some $G$, it is determined that $\mu_G \geq \beta \mu_G^0$ for some $\beta > 1$. As long as $\gamma_G^{TR} > \gamma_G^{FR}$ for every $G \in \mathcal{G}$, $\mathrm{IR}_G \geq \frac{\beta - \gamma_G^{FR}}{\gamma_G^{TR} - \gamma_G^{FR}}$.*

*Proof of Proposition 3.4.* Recall that we have defined $\mu_G = \Pr[G \mid R]$, and $\mu_G^0 = \Pr[G]$ is known by Assumption 2.1. By Bayes' rule, we have $\mu_G = \Pr[G \mid R] = \frac{\Pr[G]\Pr[R|G]}{\Pr[R]} = \mu_G^0 \frac{\Pr[R|G]}{r}$, Now, let us decompose $\Pr[R \mid G]$ by "true" reports ($Y = 1$) and "false" reports ($Y = 0$). By the law of total probability, $\Pr[R \mid G] = r \cdot \left( \gamma_G^{TR} \mathrm{IR}_G + \gamma_G^{FR}(1 - \mathrm{IR}_G) \right)$; more precisely,

$$\frac{1}{r} \Pr[R \mid G] = \Pr[R \mid G, Y = 1] \Pr[Y \mid G] + \Pr[R \mid G, Y = 0](1 - \Pr[Y \mid G])$$
$$= \gamma_G^{TR} \mathrm{IR}_G + \gamma_G^{FR}(1 - \mathrm{IR}_G)$$
$$= \gamma_G^{FR} + \mathrm{IR}_G(\gamma_G^{TR} - \gamma_G^{FR});$$

combining this with the Bayes' rule computation, cancelling the $\frac{1}{r}$ factor, gives us $\mathrm{IR}_G = \frac{\frac{\mu_G}{\mu_G^0} - \gamma_G^{FR}}{\gamma_G^{TR} - \gamma_G^{FR}}$. The result follows from the assumption that $\mu_G/\mu_G^0 \geq \beta$. □

## C.2. Omitted proofs for Sequential Z-test

We prove Theorem 4.1, restated below.

**Theorem 4.1** (Validity). *Running Algorithm 1 with $\theta_t(\alpha)$ as in Equation (3), setting $C = 1/2$, and $\omega_t^G$ updated as in Equation (2), guarantees that the probability that $\mathcal{G}^{Flag}$ will ever contain a group $G$ where $\mathcal{H}_0^G$ is true is at most $\alpha$, i.e.*

$$\Pr\left[\exists t : \exists G \in \mathcal{G}^{Flag} \text{ s.t. } \mathcal{H}_0^G \text{ holds}\right] \leq \alpha.$$

To prove this result, we will use a foundational result known as Ville's inequality (Ville, 1939).

**Theorem C.1** (Ville's inequality). *Let $\{M_t\}_{t \in \mathbb{N}^+}$ be a non-negative supermartingale, i.e. for all $t$, $M_t \geq 0$, and $\mathbb{E}[M_{t+1} \mid \mathcal{F}_t] \leq M_t$, where $\mathcal{F}_t$ is the filtration (history) of all realizations of randomness up to and including time $t$. Then, for any $x \in (0, 1)$, we have $\Pr[\exists t : M_t > \mathbb{E}[M_0]/x] \leq x$.*

The central thrust of our proof of Theorem 4.1 is due to Koolen (2017) (which itself draws from Balsubramani (2014)), and is a refinement of Jamieson et al. (2014)); we reproduce the argument in the context of our work below, though we emphasize that we do not claim the proof technique as ours.

*Proof of Theorem 4.1.* It is sufficient to show that for any group $G$ where $\mathcal{H}_0^G$ holds, we have $\Pr[\exists t : G \in \mathcal{G}^{Flag}] \leq \alpha/|\mathcal{G}|$; the statement of the theorem follows from the Bonferroni correction over all $|\mathcal{G}|$ hypotheses.

Ville's inequality (Theorem C.1) appears similar in form to the statement we hope to prove; we therefore seek to transform our test statistic $\omega_t^G = \sum_{s \in [t]} \mathbf{1}[X_s \in G]$ into a quantity that can be interpreted as a (non-negative) supermartingale. Although $\{\omega_t^G\}_{t \in \mathbb{N}^+}$ is by itself clearly not a non-negative supermartingale, each $\omega_t^G$ is the sum of $t$ Bernoulli trials with mean $\mu_G$, and Bernoulli random variables are sub-Gaussian with variance parameter $1/4$. Each $\omega_t^G$ therefore satisfies the property that $\mathbb{E}[\exp(\eta(\omega_t^G - \mathbb{E}[\omega_t^G]))] \leq \exp(\eta^2/8)$.

This holds for any $\eta$, so we will construct a distribution $\phi$ on $\eta$ and use it to construct a martingale $M_t$. In particular, note that under $\mathcal{H}_0^G$, $\mathbb{E}[\omega_t^G] < t \cdot \beta\mu_G^0$. Thus, we let $S_t := \omega_t^G - \mathbb{E}[\omega_t^G] = \omega_t^G - t\beta\mu_G^0$. We will let $M_t = \int \phi(\eta) \exp(\eta S_t - t\eta^2/8)d\eta$. Then, for any distribution $\phi$, $\{M_t\}_{t \in \mathbb{N}^+}$ is a non-negative supermartingale with respect to the randomness in realizations of reports $X_t$. To see this, we have

$$\mathbb{E}[M_{t+1} \mid \mathcal{F}_t] = \mathbb{E}\left[\int \phi(\eta) \exp\left(\eta(S_t + \mathbf{1}[X_{t+1} \in G] - \beta\mu_G^0) - \frac{(t+1)\eta^2}{8}\right) d\eta \,\middle|\, \mathcal{F}_t\right]$$

$$= \int \phi(\eta) \exp\left(\eta S_t - \frac{t\eta^2}{8}\right) \mathbb{E}\left[\exp\left(\eta(\mathbf{1}[X_{t+1} \in G] - \beta\mu_G^0) - \frac{(t+1)\eta^2}{8}\right) \,\middle|\, \mathcal{F}_t\right] d\eta$$

$$\leq \int \phi(\eta) \exp\left(\eta S_t - \frac{t\eta^2}{8}\right) d\eta$$

$$= M_t,$$

where the inequality is due to $\frac{1}{t}\mathbb{E}[\omega_t^G] \leq \beta\mu_G^0$ and subgaussianity. It thus remains to use this martingale to compute an appropriate threshold $\theta_t(\alpha)$ on $\omega_t^G$.

$M_t$ will satisfy the conditions of Theorem C.1 for any choice of $\phi$, including one which puts point mass of 1 on $\eta = \eta'$ and 0 elsewhere, i.e. $\phi(\eta') = 1$ and $\phi(\eta) = 0$ for any $\eta \neq \eta'$. One path towards establishing the threshold $\theta_t(\alpha)$ is to simply pick one value of $\eta$; however, such an $\eta$ cannot depend on $t$ and would thus result in a suboptimal threshold. Instead, we will construct $\phi$ such that it is a discrete distribution, indexed by $i \in \mathbb{N}^+$, so that $\eta$ takes values $\eta_1, \ldots, \eta_i$ with probability $\phi_1, \ldots, \phi_i$; this allows each $\eta_i$ to depend on $t$ and therefore more finer-grained optimization of the threshold. Before committing to the exact distribution $\phi$, we first illustrate how $\phi_i$ and $\eta_i$ will be used in the threshold.

Note that $M_t = \sum_{i \in \mathbb{N}^+} \phi_i \exp(\eta_i S_t - t\eta_i^2/8) \geq \max_i \phi_i \exp(\eta_i S_t - t\eta_i^2/8)$, so for any $\delta$, we have

$$\{M_t \geq 1/\delta\} \supseteq \{\max_i \phi_i \exp(\eta_i S_t - t\eta_i^2/8) > 1/\delta\} = \left\{S_t \geq \min_i \left(\frac{t\eta_i}{8} + \frac{1}{\eta_i}\ln\frac{1}{\phi_i\delta}\right)\right\},$$

and thus, picking $\theta_t(\alpha) = t\beta\mu_G^0 + \min_i \left(\frac{t\eta_i}{8} + \frac{1}{\eta_i}\ln\frac{1}{\phi_i\alpha/|\mathcal{G}|}\right)$ would guarantee that $\Pr[\exists t : \omega_t^G > \theta_t(\alpha)] \leq \alpha/|\mathcal{G}|$.

Finally, we must commit to $\phi_i$, $\eta_i$, then optimize for $i$. Let $\phi_i = \frac{1}{i(i+1)}$ (note that $\sum_i \phi_i = 1$, so this is a valid distribution), $\eta_i = 2\sqrt{\frac{2\ln(1/\phi_i(\alpha/|\mathcal{G}|))}{2^i}}$, and $i = \lfloor \log_2(t) \rfloor$. For $i = \log_2(t)$ (without rounding), this would have yielded $\eta_i = 2\sqrt{\frac{2\ln((\log_2(t)+1)(\log_2(t))/(\alpha/|\mathcal{G}|))}{t}}$ and $\theta_t(\alpha) = \frac{1}{2}\sqrt{2t\ln((\log_2 t)(\log_2 t + 1)/\alpha/|\mathcal{G}|)}$. The statement follows from handling the numerical impact of rounding. $\qquad\square$

*Remark* C.2. A key constant in the proof of the version of the algorithm that is valid in finite samples is the subgaussian variance parameter, for which we used $1/4$ (and which propagates to a multiplicative factor of $\sqrt{1/4} = 1/2$ on the threshold). This is because the variance *any* Bernoulli is at most $1/4$; however, this also motivates the choice of constant for the asymptotically-valid version of the test, which instead uses the variance parameter $\beta\mu_G^0(1 - \beta\mu_G^0)$.

We now prove the power result.

**Theorem 4.2** (Power). *Let $T$ be the stopping time of Algorithm 1 with $\theta_t(\alpha)$ as in Equation (3), $C = 1/2$, and $\omega_t^G$ as in Equation (2). Let $\Delta_{\max} = \max_{G \in \mathcal{G}} \mu_G - \beta\mu_G^0$. If $\Delta_{\max} > 0$, then $\Pr[T < \infty] = 1$. Furthermore, with probability $1 - \alpha/|\mathcal{G}|$, we have $T \leq \widetilde{\mathcal{O}}\left(\frac{\ln(|\mathcal{G}|) + \ln(1/\alpha)}{\Delta_{\max}^2}\right)$, and for any $\delta \in (0, \alpha/|\mathcal{G}|)$, we have with probability at least $1 - \delta$ that $T \leq \widetilde{\mathcal{O}}\left(\frac{\ln(1/\delta)}{\Delta_{\max}^2}\right)$.*

*Proof.* Let $G^\star := \arg\max_{G \in \mathcal{G}} \mu_G - \beta\mu_G^0$ and let $\Delta := \mu_{G^\star} - \beta\mu_{G^\star}^0$. Without loss of generality, we can consider only the test corresponding to $G^\star$ (while still testing at level $\alpha/|\mathcal{G}|$). Recall that for this instantiation of Algorithm 1, the test statistic $\omega_t^{G^\star} = \sum_{s \in [t]} \mathbf{1}[X_s \in G^\star]$ is simply the number of all reports belonging to $G^\star$ by time $t$, and that stopping time $T$ is the first time where $\omega_t^{G^\star}$ surpasses the threshold $\theta_t(\alpha)$, i.e., $T := \inf_{t \in \mathbb{N}^+} \omega_t^{G^\star} > t\beta\mu_{G^\star}^0 + \frac{1}{2}\sqrt{2.06t\ln\left(|\mathcal{G}|\frac{(2+\log_2(t))^2}{\alpha}\right)}$. For ease of notation, we will denote $C_1 := \frac{1}{2}\sqrt{2.06} = 0.718$ within this proof.

For the first claim, it is sufficient to show $\liminf_{t \to \infty} \Pr[T > t] = 0$.[7] Recall that, by our modeling, we can consider $\omega_t^{G^\star}$ to be the sum of $t$ i.i.d. Bernoulli trials with parameter $\mu_{G^\star}$. Applying Hoeffding's inequality to this sum yields for any $t$ that

$$\Pr[T > t] = \Pr\left[\omega_t^{G^\star} < t\beta\mu_{G^\star}^0 + C_1\sqrt{t \cdot \ln\left(|\mathcal{G}|\frac{(2+\log_2(t))^2}{\alpha}\right)}\right]$$

$$\leq \exp\left(-2\left(\Delta^2 t - 2\Delta C_1\sqrt{t \cdot \ln\left(|\mathcal{G}|\frac{(2+\log_2(t))^2}{\alpha}\right)}\right)\right).$$

Note that $\frac{\sqrt{t}\ln(\log_2(t))}{t} \to 0$; it can thus be seen that $\lim_{t \to \infty} \Pr[T > t] = \lim_{t \to \infty} \exp(-t) = 1$.

For the second claim, we apply Hoeffding's inequality again to see that for all $t$,

$$\Pr\left[\omega_t^{G^\star} \leq \mathbb{E}[\omega_t^{G^\star}] - C_1\sqrt{t\ln\left(\frac{(2+\log_2(t))^2}{\delta}\right)}\right] \leq \Pr\left[\omega_t^{G^\star} \leq \mathbb{E}[\omega_t^{G^\star}] - \sqrt{\frac{t}{2}\ln\left(\frac{1}{\delta}\right)}\right] \leq \delta.$$

Thus, with probability at least $1 - \delta$, for all $t$ simultaneously, $\omega_t^{G^\star} > t\mu_{G^\star} - C_1\sqrt{t\ln\left(\frac{(2+\log_2(t))^2}{\delta}\right)}$. The algorithm stops at time $t$ if and only if

$$t\mu_{G^\star} - C_1\sqrt{t\ln\left(\frac{(2+\log_2(t))^2}{\delta}\right)} > t\beta\mu_{G^\star}^0 + C_1\sqrt{t\ln\left(\frac{(2+\log_2(t))^2}{\alpha/|\mathcal{G}|}\right)}.$$

Rearranging, we have

$$\frac{t}{\left(\sqrt{\ln\left(\frac{(2+\log_2(t))^2}{\alpha/|\mathcal{G}|}\right)} + \sqrt{\ln\left(\frac{(2+\log_2(t))^2}{\delta}\right)}\right)^2} \geq \frac{C_1}{\Delta^2}.$$

---

[7]For a simple proof of this fact, see the solution to Problem 1.13 in Bertsekas & Tsitsiklis (2008).

Note that we can can upper bound the denominator of the left hand side as $\left( \sqrt{\ln\left(\frac{(2+\log_2(t))^2}{\alpha/|\mathcal{G}|}\right)} + \sqrt{\ln\left(\frac{(2+\log_2(t))^2}{\delta}\right)} \right)^2 \leq$ $4\ln\left(\frac{(2+\log_2(t))^2}{\min(\alpha/|\mathcal{G}|,\delta)}\right)$. Setting $\frac{t}{4\ln\left(\frac{(2+\log_2(t))^2}{\min(\alpha/|\mathcal{G}|,\delta)}\right)} \geq \frac{C_1}{\Delta^2}$ and rearranging gives

$$\frac{t}{1 + \ln((2 + \log_2(t))^2)} \geq \frac{4C_1 \ln(\max(\alpha/|\mathcal{G}|, 1/\delta))}{\Delta^2} \tag{6}$$

Thus, with probability $1 - \delta$, the algorithm terminates at the smallest $t$ satisfying Equation (6). The statement of the theorem follows from separating the two cases for $\delta < \alpha/|\mathcal{G}|$ and $\delta \geq \alpha/|\mathcal{G}|$, and noting that $\widetilde{O}$ notation suppresses the (negligible) log-log factor. $\qquad\square$

### C.3. Omitted proofs for betting-style algorithm

We first prove Theorem 4.3, restated for the sake of presentation.

**Theorem 4.3** (Validity). *Running Algorithm 1 with $\theta_t(\alpha) = \ln(|\mathcal{G}|/\alpha)$ and $\omega_t^G$ updated as per Equations (4) and (5) guarantees that the probability that $\mathcal{G}^{Flag}$ will ever contains a group $G$ where $\mathcal{H}_0^G$ is true is at most $\alpha$, i.e.*

$$\Pr\left[\exists t : \exists G \in \mathcal{G}^{Flag} \text{ s.t. } \mathcal{H}_0^G \text{ holds}\right] \leq \alpha.$$

*Proof.* First note that for any $G$ for which $\mathcal{H}_0^G$ holds, the sequence $\{\exp(\omega_t^G)\}_{t \geq 0}$ is a non-negative super-martingale. The non-negative property follows directly from the quantity being an exponential of a real (albeit possibly negative) number, while the fact that it is a super-martingale follows from the computations below:

$$
\begin{aligned}
\mathbb{E}[\exp(\omega_t^G)|\mathcal{F}_t] &= \mathbb{E}[\exp(\omega_{t-1}^G + \ln(1 + \lambda_t^G(\mathbf{1}_{X_t \in G} - \beta\mu_G^0)))|\mathcal{F}_t] \\
&= \exp(\omega_{t-1}^G) \cdot (1 + \lambda_t^G(\mathbb{E}[\mathbf{1}_{X_t \in G}|\mathcal{F}_t] - \beta\mu_G^0)) \\
&= \exp(\omega_{t-1}^G) \cdot (1 + \lambda_t^G(\mu_G - \beta\mu_G^0)) \\
&\leq \exp(\omega_{t-1}^G) \cdot (1 + \lambda_t^G(\beta\mu_G^0 - \beta\mu_G^0)) \\
&= \exp(\omega_{t-1}^G),
\end{aligned}
$$

where the first equality follows by Eq. 4, the second by re-arranging and noting that all quantities except $\mathbf{1}_{X_t \in G}$ are completely determined by $\mathcal{F}_t$[8], and the third by definition (see Section 2). Finally, the inequality follows because $\mu_G \leq \beta\mu_G^0$ under $\mathcal{H}_0^G$ and $\lambda_t^G \geq 0$.

Next, for any group $G$ such that $\mathcal{H}_0^G$ holds, we can apply Ville's inequality (Theorem C.1), plugging in the super-martingale $\{\exp(\omega_t^G)\}_{t \geq 0}$ and taking $x$ to be $\theta_t(\alpha) = \log(|\mathcal{G}|/\alpha)$. This yields the following guarantee:

$$
\begin{aligned}
\Pr[\exists t : \omega_t^G > \log(|\mathcal{G}|/\alpha)] = \Pr[\exists t : \exp(\omega_t^G) > |\mathcal{G}|/\alpha] \\
\leq \mathbb{E}[\exp(\omega_0^G)] \cdot \alpha/|\mathcal{G}| \\
= \alpha/|\mathcal{G}|,
\end{aligned}
$$

where the final line follows because $\omega_0^G$ is initialized as $0$ and hence $\exp(\omega_0^G)$ is equal to $1$.

Finally, by union bound we get the desired guarantee:

$$
\begin{aligned}
\Pr[\exists t : \exists G \in \mathcal{G}^{\text{Flag}} \text{ s.t. } \mathcal{H}_0^G \text{ holds}] &\leq \sum_{G \text{ s.t. } \mathcal{H}_0^G \text{ holds}} \Pr[\exists t : \omega_t^G > \log(|\mathcal{G}|/\alpha)] \\
&\leq |G \text{ s.t. } \mathcal{H}_0^G \text{ holds}| \cdot \alpha/|\mathcal{G}| \\
&\leq \alpha.
\end{aligned}
$$

$\qquad\square$

Before proving Theorem 4.4, we first state and prove some helper results.

---

[8]In particular, it is imposed that $\lambda_t^G$ be 'predictable' which precisely implies that it is fixed given $\mathcal{F}_t$.

**Claim C.3.** *For any $T \geq 4$ and group $G$, we have that the expected value over the randomness in the realizations of each $X_t$ of $\omega_T^G$ defined as per Equations (4) and (5) can be lower bounded as*

$$\mathbb{E}[\omega_T^G] \geq \mathbb{E}\left[\max_{\lambda \in [0,1]} \omega_T(\lambda)\right] - 2\ln T.$$

*where we define $\omega_T^G(\lambda)$ to be the quantity obtained by applying Equation (4) with $\lambda_t^G := \lambda$ for all $t \in [T]$.*

*Proof.* By the definition of regret we have that $\max_{\lambda \in [0,1]} \omega_T^G(\lambda) - \omega_T^G \leq R_T$. Rearranging and taking expectations, we have

$$\mathbb{E}[\omega_T^G] \geq \mathbb{E}\left[\max_{\lambda \in [0,1]} \omega_T^G(\lambda)\right] - \mathbb{E}[R_T].$$

Next, it can be verified that Equation (5) implements the Online Newton Step algorithm for $\ln(1 + \lambda_t^G(\mathbf{1}_{X_t \in G} - \beta\mu_G^0))$ (see Appendix C of Cutkosky & Orabona (2018)). We therefore have that $R_T \leq \frac{1}{2 - \ln(3)} \ln(T+1)$ in general, and $R_T \leq 2\ln(T)$ for $T \geq 4$. The statement of the claim plugging this into the expression above. $\qquad\square$

**Lemma C.4.** *For each group $G$, taking $\lambda_G^\star = \mathrm{Proj}_{[0,1]}\left[\dfrac{\mu_G - \beta\mu_G^0}{\beta\mu_G^0(1 - \beta\mu_G^0)}\right]$ maximizes expected log-wealth (at every step $t$). The resulting expected log-wealth at time $T$ (had $\lambda_G^\star$ been used at every time $t$) is equal to*

$$\mathbb{E}\left[\omega_T^G(\lambda_G^\star)\right] = T \cdot \omega_\star^G$$

*where we denote $\omega_\star^G := \mathbb{E}[\ln(1 + \lambda_G^\star(\mathbf{1}_{X_t \in G} - \beta\mu_G^0)]$ the expected one-step wealth change under the bet $\lambda_G^\star$.*

*Proof.* For a fixed $\lambda$, the log-wealth at time $T$ is given by

$$\omega_T^G(\lambda) = N_T \ln\left(1 + \lambda(1 - \beta\mu_G^0)\right) + (T - N_T) \ln\left(1 - \lambda\beta\mu_G^0\right),$$

where $N_T = \sum_{t=1}^{T} \mathbf{1}_{X_t \in G}$. Taking expectations, we have that $\mathbb{E}[N_T] = T \cdot \mu_G$ and therefore

$$\mathbb{E}\left[\omega_T^G(\lambda)\right] = T \cdot \left[\mu_G \ln\left(1 + \lambda(1 - \beta\mu_G^0)\right) + (1 - \mu_G)\ln\left(1 - \lambda\beta\mu_G^0\right)\right]. \tag{7}$$

To maximize (7), we only need to find $\lambda_G^\star \in [0,1]$ that maximizes the expressions in the square brackets. Taking the derivative we see that the function is concave, and, therefore, we can solve for $\lambda_G^\star$ by setting the derivative to $0$ and then projecting the resulting value to $[0,1]$. This yields $\lambda_G^\star = \mathrm{Proj}_{[0,1]}\left[\frac{\mu_G - \beta\mu_G^0}{\beta\mu_G^0(1-\beta\mu_G^0)}\right]$. Plugging this back into (7) we get

$$\begin{aligned}
\mathbb{E}\left[\omega_T^G(\lambda_G^\star)\right] &= T \cdot \left[\mu_G \ln\left(1 + \lambda_G^\star(1 - \beta\mu_G^0)\right) + (1 - \mu_G)\ln\left(1 - \lambda_G^\star\beta\mu_G^0\right)\right]\\
&= T \cdot \mathbb{E}[\ln(1 + \lambda_G^\star(\mathbf{1}_{X_t \in G} - \beta\mu_G^0)]\\
&:= T \cdot \omega_\star^G.
\end{aligned}$$

$\qquad\square$

*Remark* C.5. Note that we can explicitly compute

$$\omega_\star^G = \mu_G \ln\left(1 + \frac{\Delta_G}{\beta\mu_G^0(1-\mu_G)}\right) + \ln\left(1 - \frac{\Delta_G}{1 - \beta\mu_G^0}\right),$$

where $\Delta_G = \mu_G - \beta\mu_G^0$, but this quantity is difficult to analyze, and it is not clear that $\omega_\star^G$ can be explicitly lower bounded as $O(\Delta_G)$.

We now prove Theorem 4.4, which we restate below.

**Theorem 4.4** (Power)**.** *Let $T$ be the stopping time of Algorithm 1 with $\theta_t(\alpha) = \ln(|\mathcal{G}|/\alpha)$ and $\omega_t^G$ updated as per Equations (4) and (5). If $\max_{G \in \mathcal{G}} \mu_G - \beta\mu_G^0 > 0$, then, we have that $\Pr[T < \infty] = 1$. Furthermore,*

$$\mathbb{E}[T] \leq \mathcal{O}\left(\frac{1}{\omega_\star^2} + \frac{\ln(|\mathcal{G}|) + \ln(1/\alpha)}{\omega_\star}\right)$$

*where $\omega_\star := \max_{G \in \mathcal{G}, \lambda \in [0,1]} \mathbb{E}[\ln(1 + \lambda(\mathbf{1}_{X_t \in G} - \beta\mu_G^0))]$ is the maximal expected one-step increase in $\omega_t^G$ over all groups and choices of $\lambda$.*

*Proof.* Let $G^\star := \arg\max_G \omega_\star^G$ and denote the corresponding one-step wealth change $\omega_\star = \omega_\star^{G^\star}$. Note that under the alternative this will correspond to a strictly positive quantity and is equivalent to the definition in the theorem statement. We can analyze the likelihood that its null has not been rejected by time $t$ as follows:

$$\Pr\left[\omega_t^{G^\star} < \ln(|\mathcal{G}|/\alpha)\right] = \Pr\left[\omega_t^{G^\star} - \mathbb{E}[\omega_t^{G^\star}] < \ln(|\mathcal{G}|/\alpha) - \mathbb{E}[\omega_t^{G^\star}]\right]$$
$$\leq \Pr\left[\omega_t^{G^\star} - \mathbb{E}[\omega_t^{G^\star}] < \ln(|\mathcal{G}|/\alpha) - (t \cdot \omega_\star - 2\ln t)\right],$$

where the inequality follows by Claim C.3 and Lemma C.4, and the fact that $\mathbb{E}[\max_{\lambda \in [0,1]} \omega_t^{G^\star}(\lambda)] \geq \mathbb{E}[\omega_t^{G^\star}(\lambda_\star^{G^\star})] = t \cdot \omega_\star$. Whenever $t$ is large enough such that $\frac{\ln(t)}{t} \leq \frac{\omega_\star}{4}$, we have

$$\Pr[\omega_t^{G^\star} < \ln(|\mathcal{G}|/\alpha)] \leq \Pr\left[\omega_t^{G^\star} - \mathbb{E}[\omega_t^{G^\star}] < \ln(|\mathcal{G}|/\alpha) - \tfrac{3}{4}(t \cdot \omega_\star)\right]. \tag{8}$$

Since $\sqrt{t} \geq \ln t$ for all $t \in \mathbb{N}^\star$, this is satisfied in particular by taking $t \geq \frac{2^4}{\omega_\star^2}$. Further, note that $\ln(|\mathcal{G}|/\alpha) \leq \frac{t \cdot \omega_\star}{4}$ whenever $t \geq \frac{2^2 \cdot \ln(|\mathcal{G}|/\alpha)}{\omega_\star}$. So, for $t \geq \max\{\frac{2^4}{\omega_\star^2}, \frac{2^2 \cdot \ln(|\mathcal{G}|/\alpha)}{\omega_\star}\}$, we have

$$\Pr[\omega_t^{G^\star} < \ln(|\mathcal{G}|/\alpha)] \leq \Pr\left[\omega_t^{G^\star} - \mathbb{E}[\omega_t^{G^\star}] < -\tfrac{1}{2}(t \cdot \omega_\star)\right].$$

Now, note that since $\lambda_t^G \in [0, 1]$, we have that each $\ln(1 + \lambda_t^G(\mathbf{1}_{X_t \in G} - \beta\mu_G^0))$ lies in $[\ln(1 - \beta\mu_G^0), \ln(2 - \beta\mu_G^0)]$ and is therefore sub-Gaussian with parameter $\sigma = \frac{1}{2}\ln\left(1 + \frac{1}{1 - \beta\mu_G^0}\right)$; then, Hoeffding's inequality gives

$$\Pr\left[\omega_t^{G^\star} - \mathbb{E}[\omega_t^{G^\star}] < -\tfrac{1}{2}(t \cdot \omega_\star)\right] = \Pr\left[\sum_{i \in [t]} \ln(1 + \lambda_t^{G^\star}(\mathbf{1}_{X_t \in G^\star} - \beta\mu_{G^\star}^0)) - \mathbb{E}[\omega_t^{G^\star}] \leq -\frac{1}{2}t \cdot \omega_\star\right]$$

$$\leq \exp\left(-\frac{(\frac{1}{2}t \cdot \omega_\star)^2}{\frac{1}{2}t\ln^2(1 + \frac{1}{1 - \beta\mu_{G^\star}^0})}\right)$$

$$= \exp\left(-\frac{\omega_\star^2}{2\ln^2(1 + \frac{1}{1 - \beta\mu_{G^\star}^0})} \cdot t\right)$$

$$\leq \exp\left(-\frac{(1 - \beta\mu_{G^\star}^0)^2}{2} \cdot \omega_\star^2 \cdot t\right).$$

where for the last inequality we used $\ln(1 + x) \leq x$. Now we are ready to analyze the stopping time $T$ of Algorithm 1.

**Test of power one.** Let $E_t$ be the event that we stop at time $t$, i.e. $E_t = \{\exists G \text{ such that } \omega_t^G \geq |\mathcal{G}|/\alpha\}$. We have that

$$\Pr[T = \infty] = \Pr\left[\lim_{t \to \infty} \cap_{s \leq t} \neg E_t\right]$$
$$= \lim_{t \to \infty} \Pr[\cap_{s \leq t} \neg E_t]$$
$$\leq \lim_{t \to \infty} \Pr[\neg E_t]$$
$$= \lim_{t \to \infty} \Pr[\forall G, \omega_t^G < \ln(|\mathcal{G}|/\alpha)]$$
$$\leq \lim_{t \to \infty} \Pr\left[\omega_t^{G^\star} < \ln(|\mathcal{G}|/\alpha)\right]$$
$$\leq \lim_{t \to \infty} \exp\left(-\frac{(1 - \beta\mu_G^0)^2}{2} \cdot \omega_\star^2 \cdot t\right)$$
$$= 0.$$

**Expected Stopping Time.** Since $T$ is a positive integer, we can express the expected stopping time as

$$
\begin{aligned}
\mathbb{E}[T] &= \sum_{t=1}^{\infty} \Pr[T > t] \\
&= \sum_{t=1}^{\infty} \Pr[\neg E_1 \wedge \ldots \wedge \neg E_t] \\
&\leq \sum_{t=1}^{\infty} \Pr[\neg E_t] \\
&= \sum_{t=1}^{\infty} \Pr[\forall G,\, \omega_t^G < \ln(|\mathcal{G}|/\alpha)] \\
&\leq \sum_{t=1}^{\infty} \Pr\left[\omega_t^{G^\star} < \ln(|\mathcal{G}|/\alpha)\right] \\
&\leq \max\left\{\frac{2^4}{\omega_\star^2}, \frac{2^2 \cdot \ln(|\mathcal{G}|/\alpha)}{\omega_\star}\right\} + \sum_{t=1}^{\infty} \exp\left(-\frac{(1 - \beta\mu_{G^\star}^0)^2}{2} \cdot \omega_\star^2 \cdot t\right) && (9) \\
&= \max\left\{\frac{2^4}{\omega_\star^2}, \frac{2^2 \cdot \ln(|\mathcal{G}|/\alpha)}{\omega_\star}\right\} + \frac{1}{\exp\left((1 - \beta\mu_{G^\star}^0)^2 \omega_\star^2/2\right) - 1} \\
&\leq \max\left\{\frac{2^4}{\omega_\star^2}, \frac{2^2 \cdot \ln(|\mathcal{G}|/\alpha)}{\omega_\star}\right\} + \frac{2}{(1 - \beta\mu_{G^\star}^0)^2 \omega_\star^2} && (10) \\
&\leq \mathcal{O}\left(\frac{1}{\omega_\star^2} + \frac{\ln(|\mathcal{G}|/\alpha)}{\omega_\star}\right)
\end{aligned}
$$

where (9) follows from the upper bound on $\Pr\left[\omega_t^{G^\star} < \ln(|\mathcal{G}|/\alpha)\right]$ for $t \geq \max\left\{\frac{2^4}{\omega_\star^2}, \frac{2^2 \cdot \ln(|\mathcal{G}|/\alpha)}{\omega_\star}\right\}$ derived in (8), and (10) follows from $\exp(x) \geq 1 + x$. $\qquad\square$

# D. Supplemental Material for Experimental Results

## D.1. Supplemental material for vaccine side effect experiments

**Data sources.** The Vaccine Adverse Event Reporting System (VAERS) is a national adverse event incident database for U.S.-licensed vaccines, co-managed by the Centers for Disease Control and Prevention (CDC) and the U.S. Food and Drug Administration (FDA) (Chen et al., 1994; Shimabukuro et al., 2015). The database is a combination of voluntary reports from patients that have received the vaccine, as well as mandatory reports from vaccine manufacturers and healthcare professionals. For this case study, we filter the set of publicly-available reports from VAERS to reports about the COVID-19 vaccine with a complaint keyword including "myocarditis." As for how a database administrator would have known to focus on myocarditis *a priori*, one might imagine, for example, that the series of case studies found in early 2021 raised the alarm that more systematic analysis was warranted for myocarditis in particular.

To determine per-demographic base rates, i.e. to compute $\{\mu_G^0\}_{G \in \mathcal{G}}$, we utilize VaxView, a government dataset tracking national vaccine coverage (publicly accessible here), managed by the CDC. VaxView does not track vaccination rates by granular subgroups, only providing coverage rates disaggregated by age, gender, and ethnicity separately. We thus impute the vaccination rates for intersections of subgroups (e.g., "12-17, M") by multiplying the known marginal rates (i.e., $\mu_{(12-17,M)}^0 := \mu_{(12-17)}^0 \cdot \mu_{(M)}^0$).

**Additional results: Impact of Bonferroni correction.** In Figure 2, we show the same axes as in Figure 1—number of reports to first alarm on the x-axis, vs. number of permutations in which an alarm was triggered on the y-axis—for the three algorithms at $\beta = 2$. Here, we show the impact of Bonferroni correction for multiple hypothesis testing on stopping time. As expected, the invalid version of the test, which has a lower threshold for rejecting each null, stops more quickly for all three algorithms (dashed, lighter). The difference between the invalid version and the valid version (solid, darker) is relatively minor, though the impact varies across algorithms.

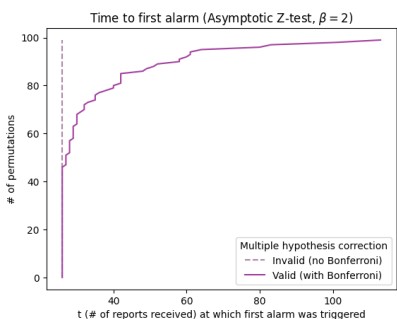 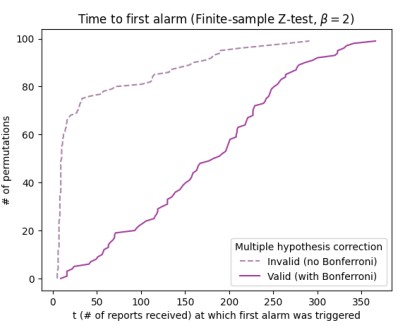 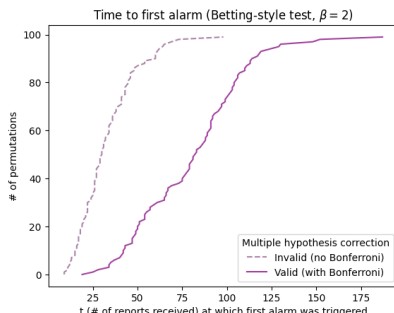

Figure 2: Impact of multiple hypothesis correction on stopping time across algorithms. As in Figure 1, each point on the plot reflects the number of trials (out of 100) in which a rejection has occurred by time $t$. In all plots, the lighter, dashed line reflects stopping time of the invalid test that does not correct for multiple testing; the dark, solid line reflects stopping time of the valid test including a Bonferroni correction.

### D.2. Supplemental material for mortgage allocation experiments

**Data sources.** We use the data (and preprocessing code) of Martinez & Kirchner (2021), which uses 2019 data from the HMDA.[9] The analysis of Martinez & Kirchner (2021) used the full year of data from 2019; we reduce the dataset to applications for conventional loans at three of the largest lending institutions, from applicants who have positive income. We assume that reports will only come from applicants whose loans were denied; in all, there are 183k applicants which satisfy these criteria.

**Reporting models.** The dataset gives several levels of financial health as measured by DTI—in ascending order, are "Struggling", "Unmanageable," "Manageable," and "Healthy." Modeling the idea that reporting behavior may be related to financial health, we use these categories to simulate the following possible patterns of reporting.

(1) *Correlated:* The likelihood of reporting increases with financial health. That is, "Healthy" denials report with probability 0.9, "Manageable" with probability 0.5, "Unmanageable" with probability 0.3, and "Struggling" with probability 0.1. Under this reporting model, the 95th-percentile (among all groups) $\rho_G/\rho$ is 1.2, and $\max_G \rho_G/\rho = 1.4$.

(2) *All Denials:* All denials submit reports regardless of financial health. Under this reporting model, the 95th-percentile $\rho_G/\rho$ is 1.5, and $\max_G \rho_G/\rho = 2.3$.

(3) *Anti-Correlated:* The (unlikely) case where individuals with worse financial health are more likely to report, i.e. "Healthy" denials report with probability 0.1, "Manageable" with probability 0.5, "Unmanageable" with probability 0.7, and "Struggling" with probability 0.9. Under this reporting model, the 95th-percentile $\rho_G/\rho$ is 1.8, and $\max_G \rho_G/\rho = 2.7$.

Note that the ground-truth ratios $\rho_G/\rho$ would have been unknown at the time that a practitioner sets $\beta$; we are able to determine these only because we have full information about the dataset and control over the reporting model. However, these computations suggest that the assumptions on reporting rates implied by the settings of $\beta = \{1.2, 1.4, 1.6, 1.8\}$ are generally reasonable, especially after considering outliers—note the disparity between the 95th-percentile vs max ratios of $\rho_G/\rho$, especially for the *All Denials* and *Anti-Correlated* models.

---

[9]The Consumer Financial Protection Bureau (CFPB) collects and publishes this data from financial institutions annually, with a two-year lag; the report (and our work) uses 2019 data which is finalized as of Dec. 31, 2022. The most recent year for which data is available is 2022, though it is available for edits through 2025.

