# OpenReview forum: "From Individual Experience to Collective Evidence: A Reporting-Based Framework for Identifying Systemic Harms"
_ICML.cc/2025/Conference — ICML 2025 poster_

### Official Review · Reviewer_zPe3 · 2025-03-13

**Overall Recommendation:** 2

**Summary:**

This paper introduces a method for identifying systemic discrimination or harm by aggregating individual reports of adverse events. The authors formalize this as the incident database problem, where reports arrive sequentially and are analyzed to detect subgroups that experience disproportionate harm.

The authors propose a sequential hypothesis testing framework that determines whether specific subgroups are overrepresented in reports of harm.

**Claims And Evidence:**

The paper proposed two statistical algorithms (Sequential Z-test and Betting-style test) that effectively identify subgroup disparities. The paper provide guarantee by derive bounds on the error probability stopping time.

**Essential References Not Discussed:**

I'm not aware of essential references that is missed.

**Experimental Designs Or Analyses:**

The experimental design in the paper is generally sound and well-structured, with strong empirical validation. I have checked real-world and simulated experiment. For the simulated experiment, it would be better to show the result from synthetic experiment with different parameters for analysis.

**Methods And Evaluation Criteria:**

The proposed method is evaluated with synthetic and real-world dataset. The paper measures how quickly the method can detect harm, which is critical for real-world applications.

**Other Comments Or Suggestions:**

NA

**Other Strengths And Weaknesses:**

- The paper tackles a well-motivated real-world problem.

- The paper is experimented with both synthetic and real-world dataset.

- The clarity and structure of the paper could be improved, for example, explicitly stating the problem definition in a more direct and structured manner.

- The evaluation of experiment seems to be limited. For example, a explicit false positive/negative rate analysis could help reader understand better.

- Assumption that baseline rates $(\mu_0 G) $ are known, which is unclear in practice.

**Questions For Authors:**

Are there other existing method that could be comparable to the proposed approach?

**Relation To Broader Scientific Literature:**

Earlier work formalize the fairness auditing as a batch hypothesis testing problem, this paper formulate the problem sequential hypothesis testing, and enables the use of existing method for sequential hypothesis testing.

**Theoretical Claims:**

I checked the proof of Prop 3.2, 3.4.

---

> ### Author Rebuttal · Authors · 2025-04-01
>
> Thanks for your time writing the review! We have grouped responses to your comments below. If there are any further weaknesses in the work that are concerning for you, please don’t hesitate to let us know.
>
> **Experiments**
>
> > _“For the simulated experiment, it would be better to show the result from synthetic experiment with different parameters for analysis.”_
>
> To clarify, the mortgage experiments in 5.2 are only partially simulated: they draw from real-world data, and the component that we simulate is reporting behavior, which is exactly the main parameter of interest for our problem setting.
>
> To this end, we simulate three different patterns of reporting, as discussed on L396. In Appendix D.2, we give more details on these simulated reports, as well as how they relate to the parameters that affect modeling (i.e. $\rho_G/\rho$) that were discussed in Section 3.1. As our computations (in D.2) show, the three ways that we simulate reports do correspond to meaningfully different values of the parameter $\rho_G/\rho$.
>
> If you meant something else by “parameter,” please let us know!
>
> > _“a explicit false positive/negative rate analysis could help reader understand better.”_
>
> In our problem setting, we felt that false positives/negatives were not quite the right abstraction for understanding the performance of algorithms. Thus, while we covered measures that are conceptually similar, we did not use the language of false positives/negatives explicitly. We will outline how these ideas relate below.
>
> For false positives: The notion of “false positive” in our setting is subtle. For the pure hypothesis testing setting, a false positive would be a group for which $\mu_G \leq \beta\mu_G^0$ but is returned by the algorithm; the likelihood of this error is what is provably controlled at level $\alpha$. For both the vaccine and mortgage experiments, all tests identified groups where $\mu_G \geq \mu_G^0$, i.e. “FPR” of 0.
>
> For our application, we are also interested in a notion of “true” harm — i.e., we hope that the groups identified by the algorithms actually reflect groups that (post-hoc) we know to have been harmed. For the vaccine experiment, this was broadly the “young men” category; all our algorithms only identified the groups (M, 12-17) and (M, 18-24), also suggesting a “FPR” of 0. For the mortgage experiment, we wanted to identify groups with a high relative risk of denial in general, but there are not necessarily hard cutoffs for what would have counted as a “true/false positive.” Tables 2-3 show that our algorithms generally found groups with high true relative risk, suggesting a low “FPR” overall.
>
> For false negatives: Because our tests are sequential and could run for arbitrarily many timesteps, it is impossible to ever fully conclude that a non-null hypothesis has not been rejected.  (In fact, our power results indicate that, in the limit of $t \to \infty$, both our proposed tests will, with probability 1, identify any group with $\mu_G > \beta\mu_G^0$.)
> Furthermore, all of our algorithms stop fairly quickly (see Tables 1, 2, 3) in almost all the settings we test. As we discuss in Table 3, there are a handful of settings where a group has not been identified within 40k steps; heuristically, these could be considered “false negatives” and we report those rates in Table 3.
>
>
> **Writing (clarity & structure; problem definition)**
>
>
> While our work does involve many moving parts, we have done our best to keep the presentation modular. Section 2 is focused fully on notation and model; the beginning of Section 3 gives the problem statement explicitly; and the beginning of Section 4 outlines our general solution concept.
>
> We would love to hear if there are specific aspects of the presentation that were unnecessarily confusing, or any concrete suggestions for revision that would improve clarity.
>
> **Knowledge of base rates**
>
> > _“Assumption that baseline rates (μ0G) are known, which is unclear in practice.”_
>
> We believe that there is a strong case to be made that it is reasonable to expect $\mu_G^0$ to be tracked by system owners.
> For example, it is already mandatory, by the Home Mortgage Disclosure Act, for banks to record and publicize the demographic details of all home loan applicants, and the CDC tracks vaccine uptake rates). Generally, we expect that most organizations track some internal metrics for system usage even if this information is not released publicly. On the other hand, we hope that for future systems, our work is one motivation to actively to ask or mandate organizations to collect/share this data.
>
> **Other methods**
>
> We considered the question of baselines for this problem carefully, but for our problem setting, the two algorithms proposed in section 4 are adaptations of the main approaches to sequential testing given in the literature. One notable alternative that we excluded was Wald’s SPRT, which requires making more parametric decisions and is thus not directly comparable.

---

### Official Review · Reviewer_y9JF · 2025-03-13

**Overall Recommendation:** 5

**Summary:**

The authors propose a framework to identify subgroups that are more likely to experience adverse events in a incident database. Therefore, they construct two algorithms that can deal with the sequentially arriving events to perform hypothesis testing. They show that their algorithms work nicely in empirical practice.

**Claims And Evidence:**

The authors break the argument for their claim (identify subgroups with adverse events leveraging reports of negative interactions) into three parts:
1) They relate reported incidence rates to true incidence rates, under assumptions on the reporting behavior of the group (proofs in appendix).
2) They show the theoretical validity and power of their two algorithms (proofs in appendix).
3) They further support it with convincing empirical evidence. They show how assumptions on the reporting behavior might be chosen and how to relax them bit by bit (while still performing valid tests).

**Essential References Not Discussed:**

No.

**Experimental Designs Or Analyses:**

The authors show two experiments, one fully empirical (real-world data and reporting) and one semi-synthetic (real-world data and synthetic reporting). They are well-conducted and give insights into expected stopping times of their algorithm.

**Methods And Evaluation Criteria:**

Their approach address how to access fairness claims from individual reports, and how to do so on a regular basis. Hence, they show a way how to implement mechanisms to ensure fair treatment of AI systems in practice.

**Other Comments Or Suggestions:**

The authors could also stress in the main paper, that it is statistically valid to rerun tests with different $\beta$. It is done so in the empirical results, but the argument that it is statistically valid is only mentioned in the appendix.

**Other Strengths And Weaknesses:**

The paper is well-written, and address an important question, especially in light of the recent political developments.

**Questions For Authors:**

1. The authors mention in the practical considerations that varying baseline preponderance can be handled under their framework. How?

**Relation To Broader Scientific Literature:**

They expand the literature on sequential testing for monitoring adverse incidents, with the focus on group fairness.

**Theoretical Claims:**

I've checked the proofs of Proposition 3.2, Proposition 3.4, Theorem 4.1 on validity of the sequential z test and Theorem 4.3 on validity of the betting style test. They are well-written and sound.

There is one tiny error in the proof of Theorem 4.1. When showing that $M$ is a supermartingale, they take the $exp(-t\eta^2/8)$ out of the conditional expectation in the second equality. However, it's also inside $exp(-(t+1)\eta^2/8)$ instead of $exp(-1\eta^2/8)$.
In my opinion, it would be worth it to show the two steps for the next inequality in detail (upperbound - $\beta\mu$ by - the expectation & subgaussianity). Showing the supermartingale property is the crucial proof step afterall, and might be checked by others too.

---

> ### Author Rebuttal · Authors · 2025-04-01
>
> Thanks for your time writing the review and reading our paper! It is a good catch on the 4.1 proof, and we agree it would be clearer to break it up as you suggested — we’ll do so in the revision!
>
> To answer your question about handling variations in $\mu_G^0$, [1] show in Section 3 how to extend their standard algorithm to handle a variety of settings where the problem varies over time. These properties come almost “for free” from the testing by betting setup, and doing something analogous for our betting-style algorithm is straightforward despite our tests themselves being different. We glossed over this point a bit in the version of the draft we submitted but will be more explicit in our revision.
>
> [1] Chugg et.al. Auditing Fairness by Betting

---

### Official Review · Reviewer_UJh6 · 2025-03-14

**Overall Recommendation:** 5

**Summary:**

This paper introduces methods for identifying subgroups disproportionately affected by AI-related harms. It does so by applying sequential hypothesis testing methods to a stream of incidents incoming into a database. Two methods are proposed: sequential Z testing and “betting-style” approach where the test essentially “bets against” the null hypothesis. The paper includes some theoretical results on validity and shows that the two algorithms are essentially equivalent from a validity perspective. The work also tests two real-world examples: myocarditis reports from COVID-19 vaccines and mortgage allocations. In both cases, they report empirical “times to first alarm” from their tests as well as relative risk metrics.

**Claims And Evidence:**

The claims in the work do seem to be supported by clear and convincing evidence. Both theoretical proofs and empirical results on real datasets are provided. I do not find any problematic claims.

**Essential References Not Discussed:**

I am not aware of any essential references that were not discussed.

**Experimental Designs Or Analyses:**

As far as I could tell without running code or exploring data myself, the experimental designs and analyses seem sound.

**Methods And Evaluation Criteria:**

The methods and evaluation criteria are appropriate for the problem being solved. The framing of the problem as a sequential hypothesis test is both a novel formulation and clever method to elucidate impacted groups.

**Other Comments Or Suggestions:**

N/A

**Other Strengths And Weaknesses:**

The primary strength of this paper is in its clever formulation of the subgroup detection problem as a hypothesis testing problem. I think this gives the method considerable flexibility to detect a wide variety of vulnerable groups. The work is clearly organized and presented, and the experimental results are convincing.

I don’t really see any major weaknesses with this work. I think it’s a well done contribution that is important and unique. It should be in ICML because it deals with the ever more important realm of AI harm detection in a novel and sound way.

**Questions For Authors:**

1. Given the times to first alarm that you observe, do you have a sense of how well these methods might perform in terms of runtime, etc. in practice in a real system? For example, could you see this being implemented on the existing AI incident database at https://incidentdatabase.ai/.

2. Do you have thoughts on how to handle a situation where you do not have access to sensitive group variables or demographic data? Would it be possible to model the group membership as a latent variable with some uncertainty? Or try to extract relevant variables from the report itself?

**Relation To Broader Scientific Literature:**

This work is an important contribution to the broader literature on AI harms. Previous work has established and described some of the incident databases of the type described in this paper. Others have taxonomized systemic risks of harm to people. None, to my knowledge, have proposed such a method for identifying groups that are more likely at risk of harm based on existing and incoming incident reports.

**Theoretical Claims:**

I did not verify the proofs in detail.

---

> ### Author Rebuttal · Authors · 2025-04-01
>
> Thanks for your time writing the review and the thoughtful questions!
>
> (Q1) We are overall optimistic about how our methods might work for a real-world system, and in future work we hope to develop and/or highlight collaborations with practitioners with real incident reporting databases. The linked AI incident database is not directly compatible with our framework (see also our discussion in Appendix A). While incidentdatabase.ai collects one-off stories of problematic incidents, they are not necessarily linked to specific systems — any incident with any AI system is eligible for report — which makes it hard to make claims about patterns of problems with specific systems. While perhaps those incident reports could be separated by an associated AI system, an additional challenge is to formalize and identify appropriate “base rates” or “subgroups” for this setting.
>
> (Q2) This is definitely a question on our minds for future work. One related work we mention [1] identifies subgroups by clustering online, but the task of running a hypothesis test is much more stringent than making predictions. Dealing with the intricacies of a sequential hypothesis test is the main technical challenge in this setting — such an approach must address data reuse for both identifying subgroups and running a valid test (i.e., avoiding the sequential analogue of p-hacking).
>
> [1] Dai et.al. Learning With Multi-Group Guarantees For Clusterable Subpopulations

---

### Official Review · Reviewer_c4AK · 2025-03-21

**Overall Recommendation:** 4

**Summary:**

This paper work studies the problem of identifying systemic harms through individual reporting mechanisms using incident databases where individuals can report negative interactions with a system (such as loan denials or vaccine side effects) to identify subgroups disproportionately experiencing harm. The authors frame this as a sequential hypothesis testing problem and for each subgroup, test whether that group is overrepresented in reports relative to their representation in the base population by a factor of β. Under their assumptions about reporting behavior, this overrepresentation serves as a proxy for actual disparities in harm. Two approaches are are looked at for operationalizing: a sequential Z-test and a betting-style approach. The authors present results on two real-world applications: identifying myocarditis risk from COVID-19 vaccines in young men, and detecting racial disparities in mortgage loan approvals. In both cases, the methods successfully identify known instances of disproportionate harm using only a fraction of the data that was originally used to discover these issues.

**Claims And Evidence:**

Yes, I think the technical claims made in this paper are well supported. The authors provide a nice formulation of the problem in terms of hypothesis testing, with Theorems 4.1-4.4 which establishing the control of false positives and power of the proposed approaches. It would seem that we may encounter difficult as the number of groups grows large, but I don't feel that is a fundamental flaw.

**Essential References Not Discussed:**

N/A

**Experimental Designs Or Analyses:**

See above re:my comment on not being a subject matter expert, but the design themselves are good and demonstrate the central claims of the paper.

**Methods And Evaluation Criteria:**

Yes, I think the evaluation is reasonable. I'm not a public health expert so I'm not sure whether the scenarios are exactly aligned with what would be used in practice, but to my reading this makes sense.

**Other Comments Or Suggestions:**

N/A

**Other Strengths And Weaknesses:**

I found this paper to be well described and implemented.  The paper creatively combines individual reporting, sequential testing, and fairness auditing into a coherent framework. This integration addresses a real gap in current practice for assessing disparate impacts. I thought the authors do a nice job of looking at multiple statistical approaches and assessing their efficacy. The empirical evidence is also quite good.

My main concern is how scalable this method would be in practice, but I believe that to be second order.

**Questions For Authors:**

1. What guarantees or assumptions are needed regarding access to the reporting system across different demographic groups? Since the method relies on comparing the proportion of reports from a group to their base population proportion, would systemic barriers to report submission (e.g., technology access, language barriers) impact the validity of your conclusions?
2. While you demonstrate effectiveness with 29 and 115 groups respectively, how does your approach scale to settings where the number of potential subgroups grows combinatorially with the number of features? Are there modifications to make this more efficient beyond the Bonferroni correction?

**Relation To Broader Scientific Literature:**

This work builds on pre-deployment auditing and batch post-hoc methods (Cen & Alur, 2024; Cherian & Candès, 2023) by creating a continuous post-deployment monitoring framework, dynamically discovering affected subgroups rather than relying solely on predefined protected categories, sharing philosophical goals with multicalibration (Hebert-Johnson et al., 2018). From a statistical perspective, the work uses sequential testing frameworks to fairness monitoring, connecting with recent applications by Chugg et al. (2024) and Feng et al. (2024) but with distinct test objectives. The betting-style algorithm leverages cutting-edge advances in sequential hypothesis testing (e.g., Waudby-Smith & Ramdas, 2024).

**Theoretical Claims:**

Yes, I read through all proofs, and they are all sound.

---

> ### Author Rebuttal · Authors · 2025-04-01
>
> Thanks for your time writing this review and the thoughtful questions!
>
> (Q1) This is a good question --- differential rates of (access to) reporting is something that we’ve thought about a lot. In the current version of this work, this can be modeled with the group-specific reporting parameters discussed on section 3 — and, though we don’t focus on it in the main exposition, in principle a different $\beta$ could be set for each group. That said, while it is natural to model known underreporting (e.g. that arises due to access reasons), the current version of our framework doesn’t help with identifying or estimating the _degree_ of underreporting (which, e.g., some of the related work on L80-86 addresses). We think better understanding this question is an important direction of future work.
>
> (Q2) Our theory shows that the stopping time increases by only a logarithmic factor in the number of groups, and only additively (rather than multiplicatively). Thus, for settings where the number of groups is combinatorially large (e.g. we have $2^d$ groups in the case of $d$ binary features), we would expect an additive impact on the stopping time of approximately $O(d)$. Our experiments with 29 and 115 groups suggest that the impact of Bonferroni in practice is even less pronounced than the $\log(|\mathcal{G}|)$ suggested by theory, and we suspect this to be true in general.
>
> As for algorithmic improvements, it is not obvious that current mathematical tools allow for any improvement over the Bonferroni correction. Some recent developments in sequential testing with e-values can handle composite null testing (e.g. [1]) — however, their guarantee is subtly different, in that they are only able to confirm that a harm _has_ occurred to one of the groups, but not identify which one it was. This is an area of future work we are definitely interested in, though it seems that it will require developing more sophisticated theory.
>
> [1] Cho et.al. Peeking with PEAK: Sequential, Nonparametric Composite Hypothesis Tests for Means of Multiple Data Streams

---

### Decision · Program_Chairs · 2025-05-01

**Decision:**

Accept (poster)

**Comment:**

This paper introduces a framework for identifying systemic harms through sequential analysis of individual reports. It formalizes the "incident database" problem using tools from sequential hypothesis testing and applies these methods to both real and semi-synthetic datasets. The proposed approach is well-motivated and addresses the growing need for post-deployment auditing methods that are both flexible and statistically grounded.

The reviewers provided generally positive evaluations. One reviewer nicely summarized that "the paper creatively combines individual reporting, sequential testing, and fairness auditing into a coherent framework." I share this view. The theoretical analysis is sound, and the experiments help illustrate the practical applicability of the framework.

One reviewer raised concerns regarding clarity and the depth of empirical evaluation. However, the authors responded thoroughly and constructively, addressing the key points and offering specific revisions. I find their responses reasonable and satisfactory.
One additional aspect that may benefit from clarification is the definition of “harm.” While the paper adopts a statistical framing—focused on subgroup overrepresentation relative to baseline rates—harm is often understood as a causal notion. A short discussion in the final version comparing this statistical notion of harm with causal definitions, and how it aligns with or differs from related literature, would help position the work more clearly.

Overall, I believe this paper offers a thoughtful and well-executed approach to fairness monitoring in practice. I recommend acceptance.